# Robust Label Proportions Learning

Jueyu Chen[1], Wantao Wen[1,*], Yeqiang Wang[2,*], Erliang Lin[1,*], Yemin Wang[3], Yuheng Jia[4,†]

[1]School of Artificial Intelligence, Southeast University, Nanjing, China
[2]Northwest A&F University, Xianyang, China
[3]School of Informatics, Xiamen University, Xiamen, China
[4]School of Computer Science and Engineering, Southeast University, Nanjing, China
{jueyuc,213211883,213223797}@seu.edu.cn, Wangyeqianger@126.com,
wangyemin@stu.xmu.edu.cn, yhjia@seu.edu.cn

## Abstract

Learning from Label Proportions (LLP) is a weakly-supervised paradigm that uses bag-level label proportions to train instance-level classifiers, offering a practical alternative to costly instance-level annotation. However, the weak supervision makes effective training challenging, and existing methods often rely on pseudo-labeling, which introduces noise. To address this, we propose RLPL, a two-stage framework. In the first stage, we use unsupervised contrastive learning to pretrain the encoder and train an auxiliary classifier with bag-level supervision. In the second stage, we introduce an LLP-OTD mechanism to refine pseudo-labels and split them into high- and low-confidence sets. These sets are then used in LLPMix to train the final classifier. Extensive experiments and ablation studies on multiple benchmarks demonstrate that RLPL achieves comparable state-of-the-art performance and effectively mitigates pseudo-label noise.

## 1 Introduction

Learning from Label Proportions (LLP), a significant weakly-supervised learning paradigm [1, 3, 36, 15, 13], addresses scenarios where individual instance labels are costly or inaccessible due to privacy concerns [10, 31]. Instead, LLP leverages more readily available label proportions within bags of instances. This practical advantage has led to LLP's application in diverse fields such as medical analysis [11], e-commerce [23], political science [27], and remote sensing [9]. The core task in LLP is to train an instance-level classifier using only these bag-level label proportions, a distinct challenge compared to traditional instance-level supervision.

The weak supervision from bag-level proportions poses significant challenges for learning accurate instance-level classifiers. As observed by Yu et al. [34], insufficient class separation in bag proportions can severely degrade performance. Many recent LLP methods therefore turn to pseudo-labeling: Ma et al. [20] re-weight high-confidence labels for an auxiliary instance loss; PLOT [19] alternates between bag-level and pseudo-label training; Liu et al. [18] employ self-ensembling. However, noisy pseudo-labels remain a major bottleneck.

We propose **R**obust **L**abel **P**roportions **L**earning (**RLPL**), a two-stage framework. In Stage 1, we pre-train an encoder via contrastive learning and train a classifier head using only bag proportions. In Stage 2, we refine the resulting pseudo-labels with LLP-OTD (LLP-penalized Optimal Transport-based Label Dividing), splitting data into a high-confidence labeled set and an unlabeled set. Finally, LLPMix, inspired by MixMatch, integrates LLP constraints into a semi-supervised pipeline to train the main classifier. Experiments on standard benchmarks show that RLPL possesses comparable performance to prior methods, and ablations confirm the effectiveness of LLP-OTD in filtering noise.

The primary contributions of this paper are:

---

*These authors contributed equally.
†Corresponding author.

39th Conference on Neural Information Processing Systems (NeurIPS 2025).

- We propose **RLPL**, a novel two-stage LLP framework that effectively leverages both bag-level proportions and instance-level pseudo-labels, demonstrating robust performance, particularly in challenging large-bag scenarios.

- We introduce **LLP-OTD**, an LLP-constrained optimal transport-based mechanism to refine noisy pseudo-labels by extracting high-confidence labels. Experiments confirm its superior performance in pseudo-label refinement.

- We develop **LLPMix**, a training mechanism that utilizes the LLP-OTD refined dataset and label proportion information within a semi-supervised framework, achieving excellent experimental results.

- Comprehensive experiments validate RLPL's comparable state-of-the-art performance against current leading LLP models, supported by thorough ablation studies. Meanwhile, our model exhibits bag-insensitive robustness.

## 2 Related Work

### 2.1 Learning from Label Proportions(LLP)

Learning from Label Proportions (LLP) is a weakly supervised paradigm in which only aggregated label proportions for predefined bags are available, and instance-level labels are inaccessible [23, 34, 12, 16, 22]. Early approaches matched predicted bag proportions to ground truth [34], but this coarse supervision admits multiple instance-label configurations, degrading classification performance. To recover finer supervision, recent methods generate pseudo-labels—for example, L2P-AHIL [20] re-weights high-confidence labels via entropy measures, and LLPFC [36] treats LLP as a label-noise problem.

Our **RLPL** framework advances LLP by introducing **LLP-OTD**, an Optimal Transport–based denoising mechanism with a post-optimization consistency heuristic to filter reliable pseudo-labels, and **LLPMix**, which treats low-confidence labels as unlabeled and enforces an explicit LLP consistency loss within a MixMatch–style semi-supervised pipeline. Together, these components yield superior instance-level classifiers from bag-level proportions.

### 2.2 Optimal Transport in LLP

Traditional LLP methods that minimize KL divergence between predicted and bag-level proportions often yield high-entropy, flat distributions lacking discriminative power [19]. To enforce more structured alignment, Optimal Transport (OT) has been introduced into LLP. Liu et al. [17] proposed OT-LLP, using an entropically regularized Sinkhorn solver to match proportions exactly, thereby boosting accuracy. La et al. [24] combined OT with prototypical contrastive learning to align embeddings with class prototypes. Liu et al. [19] also proposed PLOT (Pseudo-Label Optimization via OT), which alternates OT-based label assignment with model updates to suppress noise.

While these OT-based methods improve proportion matching, they often struggle with pseudo-label noise and reliable confidence assessment. Our **LLP-OTD** mechanism offers distinct advantages by iteratively correcting pseudo-labels via an OT process incorporating an LLP-specific penalty in its cost function, directly enforcing bag constraints during refinement. Crucially, LLP-OTD employs a novel "post-optimization consistency" metric to robustly distinguish high-confidence pseudo-labels from unreliable ones, surpassing common heuristic criteria. This focus on advanced pseudo-label correction and confidence assessment allows LLP-OTD to significantly enhance the quality of instance-level supervision for more accurate LLP classifiers.

### 2.3 Semi-Supervised Learning

**S**emi-**S**upervised **L**earning (**SSL**) aims to leverage both labeled and unlabeled data during training [37, 29, 4]. Common SSL approaches include consistency regularization, pseudo-labeling, and entropy minimization, often combined with strong data augmentations [32]. MixMatch [2] is a prominent SSL method that unifies data augmentation, label guessing with sharpening, and MixUp [35] to effectively utilize all available data. Given that the dataset refined by our **LLP-OTD** is partially labeled, resembling an SSL setting, we adapt SSL principles for LLP. However, standard SSL methods

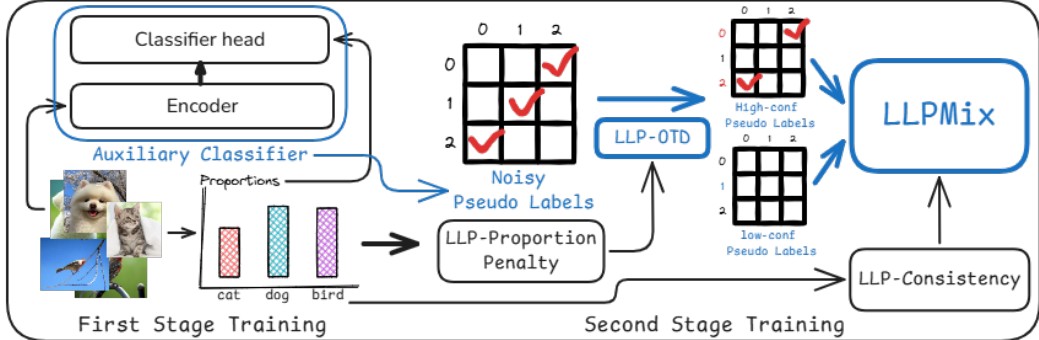

Figure 1: Overview of Methodology

like MixMatch do not inherently account for bag-level label proportion constraints crucial in LLP. Therefore, our **LLPMix** mechanism enhances a MixMatch-like framework by integrating an explicit LLP consistency loss. This ensures that the model's predictions adhere to the known bag label proportions while benefiting from the robust semi-supervised training strategy, effectively bridging SSL with the specific demands of LLP.

## 3 Preliminary: Problem Setting

In the Learning from Label Proportions (LLP) setting, we seek to train an instance-level classifier without access to individual labels. Instead, the training data $D = \{x_i\}_{i=1}^N \subset \mathcal{X}$ is partitioned into $M$ disjoint bags $B_j$, each containing $n_j$ instances. For each bag $B_j = \{x_{j,1}, \ldots, x_{j,n_j}\}$ we observe only the class proportions

$$\mathbf{p}_j = \begin{bmatrix} p_{j,1}, \ldots, p_{j,K} \end{bmatrix}^\top, \quad p_{j,k} = \frac{1}{n_j} \sum_{l=1}^{n_j} \mathbb{I}(y_{j,l} = c_k),$$

where $\mathcal{Y} = \{c_1, \ldots, c_K\}$ and $\sum_k p_{j,k} = 1$. Each instance $x_i$ has an unknown true label $y_i \in \mathcal{Y}$ and a feature embedding $f(x_i) \in \mathbb{R}^d$, produced by an encoder $f$.

Our goal is to learn a classifier $h : \mathbb{R}^d \to \Delta^K$ that, given $f(x_i)$ and the bag-level proportions $\{\mathbf{p}_j\}$, recovers accurate estimates of the hidden labels. In other words, $h \circ f$ should predict $y_i$ for each $x_i$ by leveraging only the aggregated supervision provided by the $\mathbf{p}_j$.

## 4 Method

In this section, we detail our proposed framework. Beginning with overview of the whole methodology and formal definition of problems, we give a detailed illustration of each component in our method.

### 4.1 Overview

To solve the LLP problem, we propose our **RLPL** framework as illustrated in Fig. 1. Our framework possesses two training stages. For the first stage, we utilize unsupervised contrastive representation learning strategy to train encoder and leverage bag label proportions to guide classifier head training. After first-stage training, we obtain the initial naive classifier as an auxiliary classifier to generate pseudo-labels for next-stage training. Since the initial classifier is trained under bag-level supervision, the pseudo-labels generated by this classifier imply the label proportions knowledge. These meaningful labels are sent to the second stage. However, these labels are usually noisy since the initial classifier cannot give full-correct pseudo-labeling. To cope with these noises and exploit the information brought by pseudo-labels, we propose our LLP-OTD mechanism to distinguish high-confident pseudo-labels out of the pseudo-label set and discard low-confident label set. This robust

distinguishing mechanism provides a strong support for the following training. After LLP-OTD divides the pseudo-label dataset into high-confident pseudo-label set and unlabeled set, we establish LLPMix to train on the **LLP-OTD**-divided dataset. Given the similarity between **LLP-OTD**-divided dataset and semi-supervised dataset, we organically integrate LLP constraints and MixMatch[2] learning into LLPMix, which not only make a full use of bag-level information but also leverage instance-level high-confident labels and samples themselves, providing a instance-bag dual-level supervision for main classifier training.

## 4.2    First Stage: Bag-level Supervised Naive Classifier

The primary goal of the first stage is to obtain an initial, albeit potentially naive, instance-level classifier by leveraging the available weak supervision signal, i.e., the bag-level label proportions. This stage consists of two steps: representation learning and initial classifier training.

Firstly, to effectively capture the underlying structure and semantics of the instance data, we employ an unsupervised contrastive representation learning strategy, such as SimCLR [7], to pre-train the feature encoder $f : \mathcal{X} \to \mathbb{R}^d$. This step learns instance representations $f(x_i)$ for all $x_i \in D$ without using any label or proportion information, providing a foundation for subsequent tasks.

Secondly, we train an initial classifier head, denoted as $h_{\text{init}} : \mathbb{R}^d \to \Delta^K$, where $\Delta^K$ is the $K$-dimensional probability simplex. This classifier takes the learned feature representation $f(x_i)$ as input and outputs a probability distribution over the $K$ classes, $\hat{\mathbf{q}}_i = h_{\text{init}}(f(x_i))$, where $\hat{\mathbf{q}}_i = [\hat{q}_{i,1}, \ldots, \hat{q}_{i,K}]^\top$ and $\hat{q}_{i,k}$ represents the predicted probability that instance $x_i$ belongs to class $c_k$.

The training of $h_{\text{init}}$ is guided solely by the bag-level label proportions $\{\mathbf{p}_j\}_{j=1}^M$. Specifically, for each bag $B_j$, we can estimate the predicted label proportions $\hat{\mathbf{p}}_j = [\hat{p}_{j,1}, \ldots, \hat{p}_{j,K}]^\top$ by averaging the predicted instance probabilities within the bag:

$$\hat{p}_{j,k} = \frac{1}{n_j} \sum_{x_i \in B_j} \hat{q}_{i,k} \tag{1}$$

The classifier head $h_{\text{init}}$ is then trained by minimizing the discrepancy between the predicted bag proportions $\hat{\mathbf{p}}_j$ and the true bag proportions $\mathbf{p}_j$ across all bags. A common choice for the loss function is the Kullback-Leibler (KL) divergence:

$$\mathcal{L}_{\text{stage1}} = \frac{1}{M} \sum_{j=1}^M D_{KL}(\mathbf{p}_j \| \hat{\mathbf{p}}_j) \tag{2}$$

Minimizing this loss encourages the classifier's average predictions within each bag to align with the known ground-truth proportions.

After training, this initial classifier $h_{\text{init}}$ can be used to generate initial pseudo-labels $\hat{y}_i$ for each instance $x_i$, typically by selecting the class with the highest predicted probability:

$$\hat{y}_i = \arg \max_{c_k \in \mathcal{Y}} \hat{q}_{i,k} \tag{3}$$

This initial stage provides pseudo-labels $\{\hat{y}_i\}_{i=1}^N$ that incorporate LLP prior knowledge by respecting bag proportions. However, due to the inherently ambiguous nature of bag-level supervision, the classifier $h_{\text{init}}$ is trained with weak signals, potentially leading to unreliable predictions and noisy pseudo-labels. Consequently, these initial pseudo-labels require further refinement and robust handling in a subsequent stage to effectively train an accurate instance-level classifier, motivating the introduction of our second stage (detailed in Section 4.3).

## 4.3    Second stage: Instance-Bag Dual-level Guided Robust Main Classifier

The initial classifier uses only bag-level supervision, but as shown by Yu et al. [34], even perfect alignment with bag proportions does not ensure correct instance-level predictions. Consequently, its pseudo-labels—though informed by LLP priors—remain noisy. To address this, we propose LLP-Optimal Transport Denoising (LLP-OTD), which separates initial pseudo-labels into high-confidence and low-confidence subsets, retaining only the former for training. This yields a refined, semi-labeled dataset of reliable pseudo-labels. Building on this, we introduce LLPMix—a MixMatch-inspired[2] semi-supervised framework that enforces bag-level consistency while leveraging the clean pseudo-labels to train the final instance-level classifier.

### 4.3.1 LLP-OTD: LLP-guided Optimal Transport Dividing

The LLP-OTD mechanism aims to refine the initial pseudo-labels $\{\hat{y}_i\}_{i=1}^N$ by employing an iterative optimal transport process. This process is guided by both the instance feature geometry, derived from the main classifier's encoder $f_{\text{main}}(x_i)$, and the known bag-level label proportions $\mathbf{p}_j$. LLP-OTD consists of two main steps: iterative pseudo-label refinement via OT, and confident sample partitioning.

**Iterative Pseudo-Label Refinement**    This refinement is performed in two OT passes. Let $f_{\text{main}}(x_i)$ be the feature representation of instance $x_i$ from the current main encoder. The initial pseudo-labels $\hat{y}_i$ are obtained from Eq. 3.

The first OT pass commences with the calculation of initial class barycenters. For each class $c_k \in \mathcal{Y}$, its barycenter $\boldsymbol{\mu}_k^{(0)} \in \mathbb{R}^d$ is computed as the mean of features $f_{\text{main}}(x_i)$ for instances $x_i$ initially assigned the pseudo-label $\hat{y}_i = c_k$:

$$\boldsymbol{\mu}_k^{(0)} = \frac{\sum_{i:\hat{y}_i=c_k} f_{\text{main}}(x_i)}{|\{i|\hat{y}_i = c_k\}|} \tag{4}$$

Should any class lack assigned instances, its barycenter can be initialized using random values or alternative heuristics. Subsequently, an LLP-aware cost matrix $C^{(0)}$ is constructed. The cost $C_{k,i}^{(0)}$ of associating instance $x_i$ (from bag $B_{j(i)}$ with true proportions $\mathbf{p}_{j(i)}$) with class $c_k$ (represented by $\boldsymbol{\mu}_k^{(0)}$) is defined as:

$$C_{k,i}^{(0)} = \|f_{\text{main}}(x_i) - \boldsymbol{\mu}_k^{(0)}\|_2^2 + \lambda_{\text{OTD}}(1 - p_{j(i),k}) \tag{5}$$

Here, $p_{j(i),k}$ denotes the true proportion of class $c_k$ in bag $B_{j(i)}$, and $\lambda_{\text{OTD}} \geq 0$ is a hyperparameter that balances the Euclidean feature distance against the LLP proportion penalty. This penalty term serves to discourage the assignment of an instance to a class that is known to be rare or absent within its originating bag. Embedding the LLP prior directly into the cost matrix $C_{k,i}^{(0)}$ in this manner fundamentally reshapes the matching landscape, enforcing bag-level consistency at the individual instance-prototype level rather than only at the final aggregated proportion level. With the cost matrix established, an entropy-regularized optimal transport problem is solved to find an optimal transport plan $\mathbf{T}^{(1)*} \in \mathbb{R}_{\geq 0}^{K \times N}$:

$$\mathbf{T}^{(1)*} = \arg\min_{\mathbf{T} \in \mathcal{U}(\mathbf{a},\mathbf{b})} \sum_{k=1}^K \sum_{i=1}^N T_{k,i} C_{k,i}^{(0)} - \gamma H(\mathbf{T}) \tag{6}$$

where $H(\mathbf{T}) = -\sum_{k,i} T_{k,i}(\log T_{k,i} - 1)$ is the entropy regularization, $\gamma > 0$ its strength, and $\mathcal{U}(\mathbf{a},\mathbf{b})$ represents the set of valid transport plans satisfying marginal constraints $\mathbf{a} \in \mathbb{R}^K$ and $\mathbf{b} \in \mathbb{R}^N$ (typically uniform vectors, e.g., $\mathbf{a} = \frac{1}{K}\mathbf{1}_K$ and $\mathbf{b} = \frac{1}{N}\mathbf{1}_N$). The entropy-regularized OT problem in Eq. 6 is a strictly convex optimization over a compact convex set, which guarantees a unique optimal solution $\mathbf{T}^*$ that can be efficiently found using the Sinkhorn-Knopp algorithm. A formal proof of the solution's existence, uniqueness, and algorithmic convergence, along with an analysis of the LLP-Proportion Penalty's role, is provided in Appendix. The pseudo-label for each instance $x_i$ is then updated to $\hat{y}_i^{(1)}$ by selecting the class $c_k$ that receives the maximum "mass" from $\mathbf{T}^{(1)*}$:

$$\hat{y}_i^{(1)} = \arg\max_{c_k \in \mathcal{Y}} T_{k,i}^{(1)*} \tag{7}$$

The second OT pass aims to further refine these pseudo-labels. It begins by recalculating class barycenters, $\boldsymbol{\mu}_k^{(1)}$, using the updated pseudo-labels $\hat{y}_i^{(1)}$ and the same instance features $f_{\text{main}}(x_i)$:

$$\boldsymbol{\mu}_k^{(1)} = \frac{\sum_{i:\hat{y}_i^{(1)}=c_k} f_{\text{main}}(x_i)}{\left|\{i|\hat{y}_i^{(1)} = c_k\}\right|} \tag{8}$$

A new cost matrix $C^{(1)}$ is then constructed using these refined barycenters $\boldsymbol{\mu}_k^{(1)}$, following the same formulation as Eq. 5:

$$C_{k,i}^{(1)} = \|f_{\text{main}}(x_i) - \boldsymbol{\mu}_k^{(1)}\|_2^2 + \lambda_{\text{OTD}}(1 - p_{j(i),k}) \tag{9}$$

Another entropy-regularized OT problem (analogous to Eq. 6) is solved with $C_{k,i}^{(1)}$ to yield a new transport plan $\mathbf{T}^{(2)*}$. The final OT-refined pseudo-labels, $\hat{y}_i^{\text{OT}}$, are then determined from $\mathbf{T}^{(2)*}$:

$$\hat{y}_i^{\text{OT}} = \arg\max_{c_k \in \mathcal{Y}} T_{k,i}^{(2)*} \tag{10}$$

This two-pass iterative process allows the barycenter representations and pseudo-label assignments to mutually refine each other, guided by both feature similarity and LLP constraints, thereby enhancing the quality of the pseudo-labels.

**Confident Sample Partitioning** After obtaining the final OT-refined pseudo-labels $\hat{y}_i^{\text{OT}}$, we partition the dataset $D$ into a high-confidence labeled set $\mathcal{D}_L$ and a low-confidence unlabeled set $\mathcal{D}_U$. The partitioning is based on the agreement between the initial pseudo-labels $\hat{y}_i$ (from Eq. 3) and the OT-refined pseudo-labels $\hat{y}_i^{\text{OT}}$:

- **High-Confidence Labeled Set** $\mathcal{D}_L$: Instances where the initial pseudo-label and the OT-refined pseudo-label agree are considered high-confidence. Their labels are taken as $\hat{y}_i^{\text{OT}}$.

$$\mathcal{D}_L = \{(x_i, \hat{y}_i^{\text{OT}}) | x_i \in D, \hat{y}_i^{\text{OT}} = \hat{y}_i\} \tag{11}$$

- **Low-Confidence Unlabeled Set** $\mathcal{D}_U$: Instances where the pseudo-labels disagree are considered low-confidence. These instances are treated as unlabeled in the subsequent LLPMix training stage.

$$\mathcal{D}_U = \{x_i | x_i \in D, \hat{y}_i^{\text{OT}} \neq \hat{y}_i\} \tag{12}$$

This partitioning strategy aims to select more reliable pseudo-labels for supervised training while leveraging the remaining instances in an unsupervised or semi-supervised manner, thus mitigating the impact of noise from the initial pseudo-labeling. The sets $\mathcal{D}_L$ and $\mathcal{D}_U$ are then used in the LLPMix framework.

### 4.3.2 LLPMix: Semi-Supervised Learning with LLP Consistency

Building on the high-confidence labeled set $\mathcal{D}_L$ and the unlabeled set $\mathcal{D}_U$ produced by LLP-OTD, LLPMix integrates standard semi-supervised learning with an explicit bag-level consistency constraint. First, for each example in $\mathcal{D}_U$, we generate several weak augmentations, collect the model's predictions, and sharpen their average to obtain soft pseudo-labels. Next, we mix labeled and unlabeled examples—including both their inputs and labels—using the MixUp approach, thereby creating a unified training batch that blends reliable OT-refined labels with guessed labels. Finally, we optimize a combined objective comprising a supervised classification loss on the mixed labeled data, an unsupervised consistency loss on the mixed unlabeled data, and a KL divergence–based term that ensures the model's aggregated predictions over each original bag adhere to the known bag proportions. This streamlined LLPMix procedure effectively harnesses both high-confidence pseudo-labels and bag-level supervision to drive robust instance-level learning under the LLP setting.

The core of LLPMix lies in its loss function, which combines a standard supervised cross-entropy loss $\mathcal{L}_S$ for labeled data (from $\mathcal{D}_{\text{mix}}$ originating from $\mathcal{D}_L$), an unsupervised consistency loss $\mathcal{L}_U$ for unlabeled data (from $\mathcal{D}_{\text{mix}}$ originating from $\mathcal{D}_U$), and our novel LLP consistency term $\mathcal{L}_{\text{LLP-Cons}}$:

$$\mathcal{L}_{\text{LLPMix}} = \mathcal{L}_S + w_U \mathcal{L}_U + w_{\text{LLP}} \mathcal{L}_{\text{LLP-Cons}} \tag{13}$$

where $w_U, w_{\text{LLP}}$ are weighting coefficients.

**The crucial LLP consistency term**, $\mathcal{L}_{\text{LLP-Cons}}$, ensures that the model's predictions adhere to the original bag-level proportions. This term is calculated *before* the MixUp operation. Specifically, let $\mathcal{B}_{\text{orig}}$ be the set of original instances (from $\mathcal{D}_L \cup \mathcal{D}_U$) that form the basis of the current mini-batch. For each original bag $B_j$ represented in $\mathcal{B}_{\text{orig}}$, we calculate the predicted proportion $\hat{\mathbf{p}}_j^{\text{batch}}$ by averaging the predictions $h_{\text{main}}(f_{\text{main}}(\text{Aug}(x_i)))_k$ for all instances $x_i \in \mathcal{B}_{\text{orig}}$ that originated from $B_j$. For $x_i \in \mathcal{D}_L$, $\text{Aug}(x_i)$ is $x_i$ itself; for $x_i \in \mathcal{D}_U$, $\text{Aug}(x_i)$ is one of its augmented versions used for label guessing:

$$\hat{p}_{j,k}^{\text{batch}} = \frac{1}{|\{x_i \in \mathcal{B}_{\text{orig}} | x_i \in B_j\}|} \sum_{x_i \in \mathcal{B}_{\text{orig}}, x_i \in B_j} h_{\text{main}}(f_{\text{main}}(\text{Aug}(x_i)))_k \tag{14}$$

The LLP consistency loss is then the KL divergence between these batch-wise predicted proportions and the true bag proportions $\mathbf{p}_j$:

$$\mathcal{L}_{\text{LLP-Cons}} = \frac{1}{|\mathcal{B}_{\text{bags}}|} \sum_{B_j \in \mathcal{B}_{\text{bags}}} D_{KL}(\mathbf{p}_j \| \hat{\mathbf{p}}_j^{\text{batch}}) \tag{15}$$

where $\mathcal{B}_{\text{bags}}$ is the set of unique original bags represented in $\mathcal{B}_{\text{orig}}$. This term guides the main classifier to produce instance-level predictions that, when aggregated at the bag-level from pre-MixUp samples, align with the known ground-truth proportions. This injects the LLP prior directly into the semi-supervised learning phase, complementing the instance-level signals.

## 4.4 Overall Training Algorithm

The proposed RLPL framework integrates two-stage training to leverage instance-bag dual-level information to form a robust label proportions learning model. The algorithm is summarized in the appendix and outlines the complete procedure.

# 5 Experiment

## 5.1 Experimental Setup

**Dataset** We utilized four standard benchmark datasets commonly employed in Learning from Label Proportions (LLP) research. These datasets are CIFAR-10, CIFAR-100 [14], SVHN [21], and Mini-ImageNet [30]. Both the CIFAR-10 and CIFAR-100 datasets [14] contain 50,000 training images and 10,000 test images. Each image is a $32 \times 32$ color natural scene, categorized into 10 and 100 classes, respectively. The SVHN dataset consists of $32 \times 32$ RGB images of digits, with 73,257 images for training and 26,032 for testing; additional training samples were not used in our experiments. Mini-ImageNet, a subset of the ImageNet dataset, includes 100 classes, each with 80 images for training and 20 for testing, all resized to $64 \times 64$ pixels.

**Baseline** We compare RLPL against seven representative LLP approaches. **LLPFC** formulates learning from label proportions by minimizing the KL divergence between predicted and true bag proportions within a fuzzy-clustering framework [23]. **DLLP** employs an end-to-end convolutional network that integrates labeled samples and bag-level proportions via a reshaped cross-entropy loss [25]. **LLP-VAT** augments virtual adversarial training with consistency regularization to enforce smoothness in instance predictions under local perturbations [28]. **OT-LLP** leverages entropically regularized optimal transport to impose exact proportion constraints on the classifier [17]. **SoftMatch** overcomes the quantity–quality trade-off by weighting pseudo-labels using a truncated Gaussian function combined with uniform alignment [5]. **FLMm** derives a mean-operator–based sufficient statistic for proper scoring losses, enabling learning from bag proportions without instance labels [33]. Finally, **L$^2$P-AHIL** introduces dual entropy-based weights to form auxiliary high-confidence instance-level losses, jointly optimized with bag-level supervision [20].

## 5.2 Implementation Details

**Bag Partition** For each dataset, bags of a specified size $M$ were formed by randomly selecting $M$ samples from the training set, ensuring that samples in distinct bags do not overlap. The class proportion information within each bag guided the training process, without the use of true instance labels. Following established practices [20], we selected $M$ from the set $16, 32, 64, 128, 256$. Since each dataset contains a balanced number of samples per class, this bag generation method yields relatively balanced class proportions.

**Results and Analysis** Table 1 presents classification accuracies of RLPL and state-of-the-art baselines on CIFAR-10, CIFAR-100, SVHN, and MiniImageNet under five different bag sizes (based on [20]). Across all datasets, RLPL demonstrates competitive or superior performance compared to prior methods. On MiniImageNet, RLPL achieves the best average accuracy (54.52%) and maintains a low coefficient of variation (CV=0.171), outperforming methods such as DLLP, LLP-VAT, and L$^2$P-AHIL, which exhibit significant drops as bag size increases. For CIFAR-10, RLPL achieves an average accuracy of 93.71%, slightly behind L$^2$P-AHIL (94.21%), while maintaining a very low

Table 1: Performance comparison of various methods on MiniImageNet, CIFAR-10, CIFAR-100, and SVHN datasets for different bag sizes. For each dataset, results across bag sizes [16, 32, 64, 128, 256] are reported, along with their average performance (↑) and coefficient of variation (CV, ↓). The best performing method in each column is highlighted in **bold**, and the second best is underlined. Baseline methods, excluding RLPL(Ours), are reproduced experimental results from Ma et al. [20]. The dash '-' signifies missing or inapplicable results.

| Dataset | Model | Bag Size | | | | | Average (↑) | Coeff. of Var. (↓) |
|---|---|---|---|---|---|---|---|---|
| | | 16 | 32 | 64 | 128 | 256 | | |
| MiniImageNet | LLPFC | - | - | - | - | - | - | - |
| | DLLP | 64.53 | 55.37 | 27.57 | 9.06 | 3.40 | 31.99 | 0.762 |
| | LLP-VAT | 64.17 | 54.36 | 30.96 | 9.69 | 4.90 | 32.82 | 0.717 |
| | ROT | 67.02 | 27.49 | 6.01 | 3.50 | 1.75 | 21.15 | 1.170 |
| | SoftMatch | 2.02 | 1.86 | 1.95 | 1.72 | 1.87 | 1.88 | **0.053** |
| | FLMm | - | - | - | - | - | - | - |
| | L$^2$P-AHIL | **70.26** | 59.81 | 37.51 | 16.91 | 7.46 | 38.39 | 0.627 |
| | **RLPL (Ours)** | 62.63 | **61.11** | **58.42** | **53.46** | **36.98** | **54.52** | 0.171 |
| CIFAR-10 | LLPFC | 84.10 | 71.70 | 52.71 | 20.78 | 18.79 | 49.62 | 0.531 |
| | DLLP | 91.59 | 88.61 | 79.76 | 64.95 | 44.87 | 73.96 | 0.233 |
| | LLP-VAT | 91.80 | 89.11 | 78.75 | 63.89 | 46.93 | 74.10 | 0.226 |
| | ROT | 94.86 | 94.34 | 93.97 | 92.23 | 63.10 | 87.70 | 0.141 |
| | SoftMatch | **95.24** | **95.25** | 94.23 | 93.87 | 48.20 | 85.36 | 0.218 |
| | FLMm | 92.34 | 92.00 | 91.74 | 91.54 | 91.29 | 91.78 | **0.004** |
| | L$^2$P-AHIL | 94.96 | 95.00 | **94.58** | 93.64 | 92.88 | **94.21** | 0.009 |
| | **RLPL (Ours)** | 92.54 | 94.02 | 93.50 | **94.53** | **93.95** | 93.71 | 0.007 |
| CIFAR-100 | LLPFC | - | - | - | - | - | - | - |
| | DLLP | 71.28 | 69.92 | 53.58 | 25.86 | 8.82 | 45.89 | 0.539 |
| | LLP-VAT | 73.85 | 71.62 | 65.31 | 37.36 | 2.79 | 50.19 | 0.539 |
| | ROT | 72.74 | 69.31 | 17.48 | 11.02 | 2.86 | 34.68 | 0.867 |
| | SoftMatch | **80.14** | 2.40 | 2.04 | 2.12 | 1.98 | 17.74 | 1.759 |
| | FLMm | 66.16 | 65.59 | 64.07 | 61.25 | 57.10 | 62.83 | 0.053 |
| | L$^2$P-AHIL | 78.65 | **77.30** | **76.52** | **72.21** | 23.56 | 65.65 | 0.322 |
| | **RLPL (Ours)** | 68.96 | 68.88 | 68.39 | 66.73 | **65.41** | **67.67** | **0.021** |
| SVHN | LLPFC | 93.04 | 23.26 | 21.28 | 20.54 | 19.58 | 35.54 | 0.810 |
| | DLLP | 96.90 | 96.93 | 96.64 | 95.51 | 94.34 | 96.06 | 0.010 |
| | LLP-VAT | 96.88 | 96.68 | 96.38 | 95.29 | 92.18 | 95.48 | 0.018 |
| | ROT | 95.54 | 94.78 | 96.75 | 26.00 | 12.15 | 65.04 | 0.581 |
| | SoftMatch | 22.39 | 19.68 | 19.60 | 19.64 | 19.57 | 20.18 | 0.055 |
| | FLMm | - | - | - | - | - | - | - |
| | L$^2$P-AHIL | **97.91** | **97.88** | **97.74** | **97.67** | **96.98** | **97.64** | 0.003 |
| | **RLPL (Ours)** | 94.64 | 94.83 | 95.02 | 94.92 | 95.18 | 94.92 | **0.002** |

CV of 0.007, indicating strong robustness. On the more challenging CIFAR-100 dataset, RLPL outperforms all baselines with the highest average accuracy (67.67%) and the lowest variability (CV=0.021), showcasing its resistance to label dilution. Similarly, on SVHN, RLPL achieves stable and high performance (Avg=94.92%, CV=0.002), comparable to L$^2$P-AHIL and significantly better than conventional baselines such as LLPFC and ROT. Overall, RLPL consistently ranks among the top performers across all datasets and bag sizes, demonstrating that it is less sensitive to the bag size setting. The consistently low coefficients of variation further verify RLPL's robustness, highlighting its capacity to maintain stable and reliable performance across varying weak supervision levels.

We also observe from Table 1 that RLPL's performance advantage is particularly pronounced in large-bag scenarios (e.g., bag sizes 128 and 256). We hypothesize this stems from the **degree of label ambiguity**. With smaller bags, the label proportions provide a relatively strong and unambiguous supervisory signal, allowing simpler methods to perform reasonably well. Conversely, as bag size increases, the label ambiguity escalates significantly; a single proportion vector can correspond to a

Table 2: Ablation Study

| Method | RLPL | w/o LLP-Proportion Penalty | w/o LLP-OTD | w/o LLPMix |
|---|---|---|---|---|
| **Accuracy (%)** | **93.95** | 93.36 | 84.98 | 92.33 |

Table 3: Performance comparison on the UCI Adult tabular dataset.

| Method | RLPL | ROT | L2P-AHIL |
|---|---|---|---|
| **Accuracy (%)** | **77.57** | 72.82 | 75.99 |

Table 4: Performance (%) on CIFAR-10 (bag size 256) with noisy label proportions.

| Noise Type | Gaussian (Mod.) | Gaussian (Heavy) | Uniform (Mod.) | Uniform (Heavy) |
|---|---|---|---|---|
| **Accuracy (%)** | 94.05 | 93.84 | 93.95 | 93.94 |

Table 5: Performance (%) on long-tailed CIFAR-10 (bag size 256) with varying imbalance ratios.

| Imbalance Ratio (IR) | 5 | 10 | 15 | 50 | 100 |
|---|---|---|---|---|---|
| **Accuracy (%)** | 93.55 | 91.00 | 89.72 | 81.61 | 70.28 |

vast number of potential instance-level label configurations. In these high-ambiguity settings, the robustness of our **LLP-OTD refinement process** becomes critical. Its ability to effectively denoise pseudo-labels from a highly ambiguous signal allows RLPL to excel and significantly outperform baselines, whereas other methods may struggle with the diluted supervision.

All experiments were conducted using a single NVIDIA RTX 4090 GPU with 24GB memory. For each setting, we repeated the experiment five times with different random seeds and report the mean and standard deviation of results. More detailed hyperparameter configurations are provided in the Appendix.

## 5.3 Ablation Study

As illustrated in Table 2, we conducted a series of ablation studies to validate the effectiveness of key components within our proposed RLPL model on CIFAR-10 Dataset setting bag size as 256. Removing the entire LLP-guided Optimal Transport Denoising (LLP-OTD) module (RLPL w/o LLP-OTD) results in the most significant performance drop, with accuracy decreasing from 93.95% to 84.98%. This underscores the critical role of the LLP-OTD module in refining pseudo-labels and substantially boosting model performance. When the LLP proportion penalty term is excluded from the OT cost function within the LLP-OTD module (RLPL w/o LLP-Proportion Penalty), the accuracy falls to 93.36%, demonstrating the importance of integrating true bag-level proportion information to guide the pseudo-label correction process effectively. Furthermore, omitting the subsequent LLPMix (MixMatch-based semi-supervised learning) stage (RLPL w/o LLPMix) leads to an accuracy of 92.33%, indicating that the semi-supervised learning component successfully leverages the data refined by LLP-OTD (both the reliable labeled set and the distinguished unlabeled set) to further enhance the model's generalization capabilities. Collectively, these results clearly demonstrate that the LLP proportion penalty, the core LLP-OTD refinement module, and the LLPMix strategy all contribute positively to the final performance of RLPL, with the LLP-OTD module exhibiting a particularly pronounced impact.

## 5.4 Robustness and Generalization Analysis

We conducted further experiments to evaluate RLPL's generalization beyond vision tasks and its robustness under challenging data conditions, including noisy proportions and class imbalance.

**Generalization to Tabular Data** To assess the applicability of RLPL beyond image modalities, we performed experiments on the widely-used UCI Adult tabular dataset. As shown in Table 3, RLPL

achieves an accuracy of 77.57%, outperforming strong LLP baselines. This result demonstrates that our framework is effective in the tabular data domain and generalizes well to non-vision data.

**Robustness to Noisy Proportions**    We evaluated RLPL's resilience to imperfect supervision by injecting noise into the bag proportions on CIFAR-10 (bag size 256). We applied two types of noise: **Gaussian** ($p' = \text{clip}(p + \mathcal{N}(0, \sigma^2))$) and **Uniform** ($p' = \text{clip}(p + \mathcal{U}(-r, r))$) at moderate and heavy levels. Table 4 shows that RLPL's performance remains remarkably stable, with only a minimal accuracy drop even under heavy noise (e.g., 93.84% with heavy Gaussian noise). This highlights the robustness of the LLP-OTD mechanism in handling ambiguous and noisy supervisory signals.

**Robustness to Class Imbalance**    To test performance in non-uniform data distributions, we constructed a **long-tailed** version of CIFAR-10, parameterized by the imbalance ratio (IR)—the ratio of sample sizes between the most and least frequent classes. As detailed in Table 5, while accuracy naturally degrades as the imbalance becomes more extreme, RLPL maintains strong performance, achieving 70.28% even at a severe IR of 100. This demonstrates its robustness in handling highly imbalanced class distributions.

## 6    Conclusion

This paper addresses the issue of noisy pseudo-labels in Learning from Label Proportions (LLP) by proposing the Robust Label Proportions Learning (RLPL) framework. This two-stage framework first pretrains an encoder via contrastive learning and trains an initial classifier using bag proportion information. It then introduces the core LLP-OTD (LLP-penalized Optimal Transport-based Label Dividing) mechanism to refine pseudo-labels, dividing data into a high-confidence labeled set and an unlabeled set. Finally, the LLPMix strategy, inspired by MixMatch, integrates the refined pseudo-labels and bag proportion constraints within a semi-supervised pipeline to train the main classifier. Extensive experiments on standard LLP benchmark datasets demonstrate that RLPL's performance is comparable to current state-of-the-art methods, exhibiting stronger robustness, particularly in challenging large-bag scenarios. Ablation studies also validate the effectiveness of each component, especially LLP-OTD in filtering noise. Future research directions include extending RLPL to more complex data modalities, exploring adaptive mechanisms for LLP-OTD, and theoretically investigating its noise-filtering capabilities. We will discuss the limitations of our current work in the appendix.

## Acknowledgments

This work was supported by the National Natural Science Foundation of China under Grants U24A20322 and 62576094. This work is also supported by the Big Data Computing Center of Southeast University.

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

# A Experiment

## A.1 Detailed Experiment Results

This section provides a more comprehensive presentation of our experimental findings. In comparison to the performance summary presented in the main paper, Table 6 in this appendix additionally incorporates the standard deviation (std) for each reported metric. The inclusion of these standard deviations is intended to enhance the statistical rigor of our results and to more thoroughly demonstrate the stability of the RLPL model's performance across various datasets and bag size configurations. The average performance (Avg) and coefficient of variation (CV) values reported herein, along with the newly included standard deviations, are derived from five independent experimental runs conducted. This repetition of experiments serves to validate the consistency and reliability of our model's reported performance.

Table 6: Performance comparison of various methods on MiniImageNet, CIFAR-10, CIFAR-100, and SVHN datasets for different bag sizes (mean ± std). For each dataset, results across bag sizes [16, 32, 64, 128, 256] are reported, along with their average performance (↑) and coefficient of variation (CV, ↓). The best performing method in each column is highlighted in **bold**, and the second best is underlined. The dash '-' signifies missing or inapplicable results.

| Dataset | Model | Bag Size | | | | | Average (↑) | Coeff. of Var. (↓) |
|---|---|---|---|---|---|---|---|---|
| | | 16 | 32 | 64 | 128 | 256 | | |
| MiniImageNet | LLPFC | - | - | - | - | - | - | - |
| | DLLP | $64.53_{\pm0.41}$ | $55.37_{\pm0.38}$ | $27.57_{\pm0.20}$ | $9.06_{\pm0.14}$ | $3.40_{\pm0.14}$ | 31.99 | 0.762 |
| | LLP-VAT | $64.17_{\pm0.34}$ | $54.36_{\pm0.29}$ | $30.96_{\pm0.24}$ | $9.69_{\pm0.17}$ | $4.90_{\pm0.09}$ | 32.82 | 0.717 |
| | ROT | $67.02_{\pm0.34}$ | $27.49_{\pm0.38}$ | $6.01_{\pm0.30}$ | $3.50_{\pm0.10}$ | $1.75_{\pm0.13}$ | 21.15 | 1.170 |
| | SoftMatch | $\underline{2.02}_{\pm0.23}$ | $1.86_{\pm0.24}$ | $1.95_{\pm0.20}$ | $1.72_{\pm0.33}$ | $1.87_{\pm0.25}$ | 1.88 | **0.053** |
| | FLMm | - | - | - | - | - | - | - |
| | L²P-AHIL | $\mathbf{70.26}_{\pm0.26}$ | $59.81_{\pm0.21}$ | $37.51_{\pm0.16}$ | $16.91_{\pm0.15}$ | $7.46_{\pm0.08}$ | 38.39 | 0.627 |
| | **RLPL(Ours)** | $62.63_{\pm0.15}$ | $\mathbf{61.11}_{\pm0.22}$ | $\mathbf{58.42}_{\pm0.17}$ | $\mathbf{53.46}_{\pm0.26}$ | $\mathbf{36.98}_{\pm0.21}$ | **54.52** | 0.171 |
| CIFAR-10 | LLPFC | $84.10_{\pm0.19}$ | $71.70_{\pm0.78}$ | $52.71_{\pm0.36}$ | $20.78_{\pm0.70}$ | $18.79_{\pm0.21}$ | 49.62 | 0.531 |
| | DLLP | $91.59_{\pm0.52}$ | $88.61_{\pm0.90}$ | $79.76_{\pm1.45}$ | $64.95_{\pm0.01}$ | $44.87_{\pm0.13}$ | 73.96 | 0.233 |
| | LLP-VAT | $91.80_{\pm0.08}$ | $89.11_{\pm0.22}$ | $78.75_{\pm0.46}$ | $63.89_{\pm0.19}$ | $46.93_{\pm0.71}$ | 74.10 | 0.226 |
| | ROT | $94.86_{\pm0.68}$ | $94.34_{\pm0.65}$ | $93.97_{\pm0.96}$ | $92.23_{\pm0.81}$ | $63.10_{\pm0.84}$ | 87.70 | 0.141 |
| | SoftMatch | $\mathbf{95.24}_{\pm0.12}$ | $\mathbf{95.25}_{\pm0.14}$ | $94.23_{\pm0.18}$ | $93.87_{\pm0.22}$ | $48.20_{\pm0.36}$ | 85.36 | 0.218 |
| | FLMm | 92.34 | 92.00 | 91.74 | 91.54 | 91.29 | 91.78 | **0.004** |
| | L²P-AHIL | $94.96_{\pm0.13}$ | $95.00_{\pm0.11}$ | $\mathbf{94.58}_{\pm0.21}$ | $93.64_{\pm0.20}$ | $92.88_{\pm0.53}$ | **94.21** | 0.009 |
| | **RLPL(Ours)** | $92.54_{\pm0.06}$ | $94.02_{\pm0.07}$ | $93.50_{\pm0.05}$ | $\mathbf{94.53}_{\pm0.12}$ | $\mathbf{93.95}_{\pm0.15}$ | 93.71 | 0.007 |
| CIFAR-100 | LLPFC | - | - | - | - | - | - | - |
| | DLLP | $71.28_{\pm1.56}$ | $69.92_{\pm2.86}$ | $53.58_{\pm1.60}$ | $25.86_{\pm2.15}$ | $8.82_{\pm0.94}$ | 45.89 | 0.539 |
| | LLP-VAT | $73.85_{\pm0.22}$ | $71.62_{\pm0.07}$ | $65.31_{\pm0.33}$ | $37.36_{\pm0.63}$ | $2.79_{\pm0.67}$ | 50.19 | 0.539 |
| | ROT | $72.74_{\pm0.08}$ | $69.31_{\pm0.22}$ | $17.48_{\pm0.86}$ | $11.02_{\pm0.79}$ | $2.86_{\pm0.11}$ | 34.68 | 0.867 |
| | SoftMatch | $\mathbf{80.14}_{\pm0.12}$ | $2.40_{\pm0.15}$ | $2.04_{\pm0.10}$ | $2.12_{\pm0.13}$ | $1.98_{\pm0.20}$ | 17.74 | 1.759 |
| | FLMm | 66.16 | 65.59 | 64.07 | 61.25 | 57.10 | 62.83 | 0.053 |
| | L²P-AHIL | $78.65_{\pm0.28}$ | $\mathbf{77.30}_{\pm0.50}$ | $\mathbf{76.52}_{\pm0.23}$ | $\mathbf{72.21}_{\pm0.37}$ | $23.56_{\pm2.13}$ | 65.65 | 0.322 |
| | **RLPL(Ours)** | $68.96_{\pm0.09}$ | $68.88_{\pm0.09}$ | $68.39_{\pm0.18}$ | $66.73_{\pm0.11}$ | $\mathbf{65.41}_{\pm0.15}$ | **67.67** | **0.021** |
| SVHN | LLPFC | $93.04_{\pm0.21}$ | $23.26_{\pm0.63}$ | $21.28_{\pm0.23}$ | $20.54_{\pm0.37}$ | $19.58_{\pm0.09}$ | 35.54 | 0.810 |
| | DLLP | $96.90_{\pm0.50}$ | $96.93_{\pm0.23}$ | $96.64_{\pm0.32}$ | $95.51_{\pm0.04}$ | $94.34_{\pm0.12}$ | 96.06 | 0.010 |
| | LLP-VAT | $96.88_{\pm0.03}$ | $96.68_{\pm0.01}$ | $96.38_{\pm0.10}$ | $95.29_{\pm0.17}$ | $92.18_{\pm0.29}$ | 95.48 | 0.018 |
| | ROT | $95.54_{\pm0.10}$ | $94.78_{\pm0.13}$ | $96.75_{\pm0.11}$ | $26.00_{\pm0.43}$ | $12.15_{\pm0.57}$ | 65.04 | 0.581 |
| | SoftMatch | $22.39_{\pm0.11}$ | $19.68_{\pm0.13}$ | $19.60_{\pm0.12}$ | $19.64_{\pm0.14}$ | $19.57_{\pm0.16}$ | 20.18 | 0.055 |
| | FLMm | - | - | - | - | - | - | - |
| | L²P-AHIL | $\mathbf{97.91}_{\pm0.02}$ | $\mathbf{97.88}_{\pm0.01}$ | $\mathbf{97.74}_{\pm0.06}$ | $\mathbf{97.67}_{\pm0.17}$ | $\mathbf{96.98}_{\pm0.31}$ | **97.64** | 0.003 |
| | **RLPL(Ours)** | $94.64_{\pm0.08}$ | $94.83_{\pm0.13}$ | $95.02_{\pm0.05}$ | $94.92_{\pm0.17}$ | $95.18_{\pm0.20}$ | 94.92 | **0.002** |

## A.2 Hyperparameter Configurations

To ensure the reproducibility of our experimental results, we detail the key hyperparameter configurations for our Robust Label Proportions Learning (RLPL) framework below.

**First Stage Training (Auxiliary Classifier)** In the first stage, for training the encoder via unsupervised contrastive representation learning, we employ SimCLR strategy. The encoder backbone is ResNet-18. The projection head in SimCLR consists of a single linear layer. For SimCLR training,

we utilize the Adam optimizer, setting the learning rate to $1 \times 10^{-3}$ and the weight decay to $1 \times 10^{-6}$. Subsequently, the auxiliary classifier head, also a single MLP layer, is trained using the Adam optimizer with a learning rate of $1 \times 10^{-4}$ and a weight decay of $1 \times 10^{-7}$. This stage utilizes only bag-level supervision from the bag dataset, minimizing the Kullback-Leibler (KL) divergence between the true bag proportions $p_j$ and the predicted bag proportions $\hat{p}_j$.

**Second Stage Training (Main Classifier)**   During the second stage for training the main classifier, the LLP-Proportion Penalty coefficient, denoted as $\lambda_{OTD}$, in the LLP-OTD module's cost function is set to $0.1$. This coefficient balances the Euclidean feature distance against the LLP proportion penalty. For the LLPMix training, the weight for the LLP-Consistency loss, denoted as $w_{LLP}$ in the combined objective function $\mathcal{L}_{LLPMix}$, is set to $1 \times 10^{-4}$. The main classifier is trained using the SGD optimizer. The initial learning rate is set to $0.02$, with a momentum of $0.9$ and a weight decay of $5 \times 10^{-4}$.

**Backbone Architectures for Main Classifier**   To facilitate fair comparison with baseline methods, we adopt specific backbone architectures for the main classifier across different datasets. For the CIFAR-10 and SVHN datasets, both our model and the baselines utilize WRN-28-2 as the backbone. For the CIFAR-100 dataset, WRN-28-8 is employed. On the Mini-ImageNet dataset, ResNet-18 serves as the backbone.

**Hyperparameter Sensitivity Analysis.**   To validate the robustness of our model to hyperparameter choices, we performed a sensitivity analysis for the two key coefficients: the LLP-Proportion Penalty $\lambda_{OTD}$ (Eq. 5) and the LLP-Consistency loss weight $w_{LLP}$ (Eq. 13). We evaluated various combinations on the CIFAR-10 dataset (bag size = 256), recording the accuracy at 100 epochs. The results, presented in Table 7, demonstrate that RLPL achieves high and stable accuracy (ranging from 90.82% to 91.44%) across a wide range of values for both parameters. This indicates that our framework is not overly sensitive to their specific settings, confirming its robustness.

Table 7: Hyperparameter sensitivity analysis on CIFAR-10 (bag size 256). We report accuracy (%) at 100 epochs for different combinations of $\lambda_{OTD}$ and $w_{LLP}$.

| $\lambda_{OTD}$ \ $w_{LLP}$ | 0.05 | 0.1 | 0.2 | 0.5 | 1.0 |
|---|---|---|---|---|---|
| **0.05** | 91.06% | 90.83% | 91.44% | 90.92% | 90.82% |
| **0.1** | 90.94% | 90.95% | 91.33% | 91.17% | 91.32% |
| **0.2** | 90.97% | 91.28% | 90.92% | 90.94% | 91.20% |
| **0.5** | 91.15% | 90.98% | 91.06% | 91.12% | 91.02% |

## A.3   Additional Baseline Comparisons

To further contextualize the performance of RLPL, we conducted additional experiments on CIFAR-10 comparing our method against other notable weakly-supervised frameworks, including Count Loss[26] and GLWS[6]. As shown in Table 8, RLPL consistently outperforms these baselines across various bag sizes, particularly demonstrating a significant advantage as the bag size increases.

Table 8: Performance comparison (%) against Count Loss and GLWS on CIFAR-10.

| Bag Size | Count Loss | GLWS | RLPL (Ours) |
|---|---|---|---|
| 16 | 87.5% | 85.46% | **92.54%** |
| 32 | 83.61% | 81.11% | **94.02%** |
| 64 | 68.35% | 64.64% | **93.50%** |

## A.4   Computational Cost and Scalability Analysis

We provide a detailed analysis of the computational requirements of our framework.

**Theoretical Complexity.** The computational cost of our LLP-OTD module is dominated by two steps: 1) Constructing the $K \times N$ cost matrix, which takes $O(KNd)$ time, where $K$ is classes, $N$ is instances, and $d$ is the feature dimension; 2) Solving the entropy-regularized OT problem using the Sinkhorn-Knopp algorithm for $L$ iterations, which takes $O(LKN)$ time. The total complexity is therefore $O(KN(d + L))$. As this complexity is **linear** with respect to the number of instances $N$, our method is highly scalable and efficient for large-scale LLP problems, distinguishing it from classical OT solvers that can have $O(N^3)$ complexity.

**Impact of Sinkhorn Iterations.** The complexity depends on the number of Sinkhorn iterations, $L$. We evaluated its impact on accuracy on CIFAR-10 (bag size=256) at epoch 50. As shown in Table 9, performance saturates quickly; a small number of iterations (e.g., $L = 5$) is sufficient to achieve strong results. This confirms that $L$ can be treated as a small constant, keeping the practical cost low.

Table 9: Accuracy (%) vs. Number of Sinkhorn Iterations ($L$) on CIFAR-10 (bag size=256).

| **Iterations ($L$)** | 1 | 2 | 3 | 5 | 10 | 20 | 50 |
|---|---|---|---|---|---|---|---|
| **Accuracy (%)** | 89.24 | 89.48 | 89.96 | 90.00 | 89.61 | 89.85 | 89.68 |

**Empirical Training Time.** We profiled the single-epoch training time of RLPL against baselines on CIFAR-10 (bag size=256) on an NVIDIA RTX 4090. As shown in Table 10, RLPL's runtime is comparable to other methods like LLP-VAT, and the performance gains justify the modest increase over methods like L2P-AHIL. We also analyzed the prohibitive cost of replacing our $\mathcal{L}_{stage1}$ KL divergence with a Count Loss, which involves a dynamic programming step with $O(n_j^2)$ complexity per bag. As shown in Table 11, the training time for Count Loss scales quadratically with bag size, becoming intractable, whereas our KL divergence loss remains highly efficient.

Table 10: Single-epoch training time (seconds) on CIFAR-10 (bag size=256).

| **Model** | L2P-AHIL | ROT | LLP-VAT | **RLPL (Ours)** |
|---|---|---|---|---|
| **Time (s)** | 10.61 | 10.89 | 16.09 | 16.11 |

Table 11: Training time (s/epoch) of Stage 1 loss: KL Divergence vs. Count Loss.

| **Bag Size** | 16 | 32 | 64 | 128 |
|---|---|---|---|---|
| **Count Loss (s/epoch)** | 2245 | 5012 | 9215 | 19964 |
| **KL Divergence (s/epoch)** | 17 | 9 | 6 | 6 |

### A.5 Adherence to Bag Proportions

To quantitatively verify that our final classifier respects the original bag-level constraints, we measured the Mean Absolute Error (MAE) between the classifier's aggregated instance-level predictions and the ground-truth bag proportions. A lower MAE indicates better adherence. We compared RLPL against the strong L2P-AHIL baseline on CIFAR-10 across all bag sizes. As shown in Table 12, RLPL consistently achieves a lower MAE, demonstrating superior adherence to the bag-level supervision. This effect is particularly notable in more challenging small-bag scenarios (e.g., 30.6% lower MAE than L2P-AHIL at bag size 16), confirming that the $\mathcal{L}_{\text{LLP-Cons}}$ term in our LLPMix stage (Eq. 15) effectively preserves the proportion constraints throughout the semi-supervised training phase.

## B Theoretical Guarantees for LLP-OTD

In the main paper, we stated that the entropy-regularized optimal transport (OT) problem at the core of LLP-OTD is well-posed and efficiently solvable. Here, we provide the formal theoretical guarantees, addressing the existence and uniqueness of the solution, the convergence of the algorithm, and the role of our LLP-Proportion Penalty.

Table 12: Mean Absolute Error (MAE) (lower is better) of predicted bag proportions on CIFAR-10.

| Method | Bag Size 16 | Bag Size 32 | Bag Size 64 | Bag Size 128 | Bag Size 256 |
|---|---|---|---|---|---|
| L2P-AHIL | 0.015280 | 0.013498 | 0.010868 | 0.008544 | 0.006758 |
| **RLPL** | **0.010600** | **0.009505** | **0.008061** | **0.006725** | **0.005273** |

## B.1 Existence and Uniqueness of the Optimal Solution

We first establish that the core optimization problem in our LLP-OTD mechanism has a unique global minimizer. The problem is defined as:

$$\min_{\mathbf{T} \in \mathcal{U}(\mathbf{a},\mathbf{b})} \langle \mathbf{C}, \mathbf{T} \rangle - \gamma H(\mathbf{T}), \tag{16}$$

where $\langle \mathbf{C}, \mathbf{T} \rangle = \sum_{k,i} C_{k,i} T_{k,i}$ is the transport cost, $H(\mathbf{T}) = -\sum_{k,i} T_{k,i}(\log T_{k,i} - 1)$ is the entropy term, $\gamma > 0$ is the regularization strength, and $\mathcal{U}(\mathbf{a}, \mathbf{b})$ is the transport polytope defined by marginal constraints (as defined in the main text).

**Proof:** Let $F(\mathbf{T}) = \langle \mathbf{C}, \mathbf{T} \rangle - \gamma H(\mathbf{T})$. We prove the strict convexity of the objective function.

1. The term $\langle \mathbf{C}, \mathbf{T} \rangle$ is linear in $\mathbf{T}$, and hence convex.
2. For the negative entropy term $-H(\mathbf{T}) = \sum_{k,i} T_{k,i}(\log T_{k,i} - 1)$, its Hessian matrix is diagonal with entries $\frac{\partial^2(-H)}{\partial T_{k,i}^2} = \frac{1}{T_{k,i}}$. Since $T_{k,i} > 0$ within the domain, the Hessian is positive definite, implying $-H(\mathbf{T})$ is strictly convex.
3. Therefore, $F(\mathbf{T}) = \langle \mathbf{C}, \mathbf{T} \rangle + \gamma(-H(\mathbf{T}))$ (with $\gamma > 0$) is the sum of a convex function and a strictly convex function, and is thus convex.

Furthermore, the feasible set $\mathcal{U}(\mathbf{a}, \mathbf{b}) = \left\{ \mathbf{T} \in \mathbb{R}_{\geq 0}^{K \times N} \,\middle|\, \sum_i T_{k,i} = a_k, \sum_k T_{k,i} = b_i \right\}$ is defined by linear equalities and non-negativity constraints, making it a convex polytope. Since all constraints are closed (equalities or non-strict inequalities) and the marginals sum to 1 (i.e., $0 \leq T_{k,i} \leq 1$), $\mathcal{U}(\mathbf{a}, \mathbf{b})$ is a non-empty, compact, and convex set.

By standard results in convex optimization, a convex function $F(\mathbf{T})$ optimized over a non-empty, compact, and convex set $\mathcal{U}(\mathbf{a}, \mathbf{b})$ has a unique global minimizer $\mathbf{T}^*$.

## B.2 Convergence of the Sinkhorn-Knopp Algorithm

Next, we demonstrate that the Sinkhorn-Knopp algorithm, used to solve the OT problem, converges to the unique optimal solution $\mathbf{T}^*$. The optimal transport plan $\mathbf{T}^*$ is known to have a specific structure:

$$T_{k,i}^* = u_k \cdot K_{k,i} \cdot v_i, \quad \text{where } K_{k,i} = \exp\left(-\frac{C_{k,i}}{\gamma}\right) \tag{17}$$

The Sinkhorn-Knopp algorithm is an iterative procedure to find the scaling vectors $\mathbf{u} \in \mathbb{R}^K$ and $\mathbf{v} \in \mathbb{R}^N$ that satisfy the marginal constraints.

**Proof:** The convergence can be shown by interpreting the algorithm as an alternating projection procedure using the Kullback-Leibler (KL) divergence (a specific type of Bregman divergence). Let the sets of matrices satisfying the row and column constraints be:

- $\mathcal{C}_1 = \{\mathbf{T} \in \mathbb{R}_{\geq 0}^{K \times N} \mid \sum_i T_{k,i} = a_k\}$ (Row constraints)
- $\mathcal{C}_2 = \{\mathbf{T} \in \mathbb{R}_{\geq 0}^{K \times N} \mid \sum_k T_{k,i} = b_i\}$ (Column constraints)

Each iteration of the Sinkhorn-Knopp algorithm (which corresponds to alternating updates of $\mathbf{u}$ and $\mathbf{v}$) can be viewed as performing the following alternating KL projections:

$$\mathbf{T}^{(l+\frac{1}{2})} = \arg\min_{\mathbf{T} \in \mathcal{C}_1} \text{KL}(\mathbf{T} \,\|\, \mathbf{T}^{(l)}), \tag{18}$$

$$\mathbf{T}^{(l+1)} = \arg\min_{\mathbf{T} \in \mathcal{C}_2} \text{KL}(\mathbf{T} \,\|\, \mathbf{T}^{(l+\frac{1}{2})}), \tag{19}$$

Since $\mathcal{C}_1$ and $\mathcal{C}_2$ are convex sets, and their intersection $\mathcal{C}_1 \cap \mathcal{C}_2 = \mathcal{U}(\mathbf{a}, \mathbf{b})$ is non-empty (containing the unique optimal solution $\mathbf{T}^*$), the theory of alternating Bregman projections guarantees that this iterative process converges [8]. Therefore, the sequence of iterates $\mathbf{T}^{(l)}$ produced by the Sinkhorn-Knopp algorithm is guaranteed to converge to the unique optimal solution $\mathbf{T}^*$.

### B.3 Role of the LLP-Proportion Penalty

Finally, we provide the intuition for why our innovative LLP-Proportion Penalty helps align the pseudo-label distribution with the true LLP prior. Our effective cost function (Eq. 5 in the main paper) is:

$$C_{k,i} = \|f_{\text{main}}(x_i) - \boldsymbol{\mu}_k\|_2^2 + \lambda_{\text{OTD}}(1 - p_{j(i),k}) \tag{20}$$

The key component is the penalty term $\lambda_{\text{OTD}}(1 - p_{j(i),k})$. To understand its effect, we analyze the part of the objective function corresponding to this penalty, which we denote $\mathcal{L}_{\text{LLP-Penalty}}(\mathbf{T})$:

$$\mathcal{L}_{\text{LLP-Penalty}}(\mathbf{T}) := \sum_{k=1}^{K} \sum_{i=1}^{N} T_{k,i} \left[ \lambda_{\text{OTD}}(1 - p_{j(i),k}) \right] \tag{21}$$

Grouping by bags $B_j$, this is equivalent to:

$$\mathcal{L}_{\text{LLP-Penalty}}(\mathbf{T}) = \lambda_{\text{OTD}} \sum_{j=1}^{M} \sum_{i \in B_j} \sum_{k=1}^{K} T_{k,i}(1 - p_{j,k}) \tag{22}$$

Minimizing this term is equivalent to maximizing the alignment between the transport plan's implied label proportions and the true bag proportions. Let $P_T(k \mid B_j) = \sum_{i \in B_j} T_{k,i}$ be the total mass assigned to class $k$ from bag $B_j$ by the transport plan $\mathbf{T}$. (Note: this is not a normalized distribution yet, but proportional to it). Minimizing $\mathcal{L}_{\text{LLP-Penalty}}(\mathbf{T})$ is equivalent to:

$$\min_{\mathbf{T}} \sum_{j=1}^{M} \sum_{k=1}^{K} \left( \sum_{i \in B_j} T_{k,i} \right) (1 - p_{j,k}) = \min_{\mathbf{T}} \sum_{j=1}^{M} \sum_{k=1}^{K} (P_T(k \mid B_j) - P_T(k \mid B_j) \cdot p_{j,k}) \tag{23}$$

$$= \min_{\mathbf{T}} \sum_{j=1}^{M} \left( \sum_{k=1}^{K} P_T(k \mid B_j) - \sum_{k=1}^{K} P_T(k \mid B_j) \cdot p_{j,k} \right) \tag{24}$$

Since $\sum_{k=1}^{K} P_T(k \mid B_j) = \sum_{i \in B_j} \sum_{k=1}^{K} T_{k,i} = \sum_{i \in B_j} b_i$ (where $b_i$ is the marginal for instance $i$, typically $1/N$), the first term $\sum_{k=1}^{K} P_T(k \mid B_j)$ is constant with respect to the distribution of assignments within the bag. Therefore, minimizing the penalty term becomes equivalent to:

$$\max_{\mathbf{T}} \sum_{j=1}^{M} \sum_{k=1}^{K} P_T(k \mid B_j) \cdot p_{j,k} \propto \max_{\mathbf{T}} \sum_{j} \langle \mathbf{P}_T(\cdot \mid B_j), \mathbf{p}(\cdot \mid B_j) \rangle \tag{25}$$

where $\mathbf{P}_T(\cdot \mid B_j)$ represents the vector of implied proportions from bag $B_j$ derived from $\mathbf{T}$. Maximizing this dot product encourages the transport plan's implied distribution $\mathbf{P}_T$ to align with the true prior $\mathbf{p}$, which is analogous to minimizing statistical divergences like the KL divergence.

Therefore, by introducing the LLP-Proportion Penalty into the cost matrix, we directly enforce that the optimal transport plan $\mathbf{T}^*$—the unique solution to the optimization problem—favors assignments that are consistent with the known bag-level supervision.

## C Algorithm

Our overall algorithm is summarized in Algorithm 1, where the notations and definitions used are consistent with those introduced in the main paper.

---

**Algorithm 1** RLPL: Robust Label Proportions Learning

---

1: **Input:** Dataset $D = \{(x_i, B_{j(i)})\}_{i=1}^{N}$, Bag proportions $\{\mathbf{p}_j\}_{j=1}^{M}$, Number of classes $K$.

2: **Output:** Trained main classifier $h_{\text{main}}(f_{\text{main}}(\cdot))$.

3: **procedure** STAGE 1: INITIAL NAIVE CLASSIFIER

4:     Pre-train feature encoder $f : \mathcal{X} \to \mathbb{R}^d$ using SimCLR on $D$.

5:     Initialize classifier head $h_{\text{init}} : \mathbb{R}^d \to \Delta^K$.

6:     **while** not converged **do**

7:         For each instance $x_i$, compute predicted probability distribution $\hat{\mathbf{q}}_i = h_{\text{init}}(f(x_i))$.

8:         For each bag $B_j$, compute predicted proportions $\hat{p}_{j,k} = \frac{1}{n_j} \sum_{x_l \in B_j} \hat{q}_{l,k}$.

9:         Update $h_{\text{init}}$ by minimizing $\mathcal{L}_{\text{stage1}} = \frac{1}{M} \sum_{j=1}^{M} D_{KL}(\mathbf{p}_j || \hat{\mathbf{p}}_j)$.

10:     **end while**

11:     Generate initial pseudo-labels $\{\hat{y}_i\}_{i=1}^{N}$ using $\hat{y}_i = \arg\max_{c_k \in \mathcal{Y}} \hat{q}_{i,k}$.

12: **end procedure**

13: **procedure** STAGE 2: MAIN CLASSIFIER TRAINING

14:     Initialize main encoder $f_{\text{main}}$ (e.g., with weights from $f$) and main classifier head $h_{\text{main}}$.

15:     **LLP-OTD: Pseudo-Label Denoising and Partitioning**

16:     **for** each training epoch $e = 1, \ldots, E_{\text{main}}$ **do**

17:         For each instance $x_i$, extract features $f_{\text{main}}(x_i)$.

18:         Compute initial class barycenters $\boldsymbol{\mu}_k^{(0)} = \frac{\sum_{i:\hat{y}_i = c_k} f_{\text{main}}(x_i)}{|\{i | \hat{y}_i = c_k\}|}$.

19:         Construct cost matrix $C_{k,i}^{(0)} = ||f_{\text{main}}(x_i) - \boldsymbol{\mu}_k^{(0)}||_2^2 + \lambda_{OTD}(1 - p_{j(i),k})$.

20:         Solve OT problem $T^{(1)*} = \arg\min_{T \in \mathcal{U}(a,b)} \sum_{k,i} T_{k,i} C_{k,i}^{(0)} - \gamma H(T)$ to get $\mathbf{T}^{(1)*}$.

21:         Update pseudo-labels to $\hat{y}_i^{(1)} = \arg\max_{c_k \in \mathcal{Y}} T_{k,i}^{(1)*}$.

22:         Compute refined class barycenters $\boldsymbol{\mu}_k^{(1)} = \frac{\sum_{i:\hat{y}_i^{(1)} = c_k} f_{\text{main}}(x_i)}{|\{i | \hat{y}_i^{(1)} = c_k\}|}$.

23:         Construct cost matrix $C_{k,i}^{(1)} = ||f_{\text{main}}(x_i) - \boldsymbol{\mu}_k^{(1)}||_2^2 + \lambda_{OTD}(1 - p_{j(i),k})$.

24:         Solve OT using $C^{(1)}$ to get $\mathbf{T}^{(2)*}$, then final OT-refined pseudo-labels $\hat{y}_i^{\text{OT}} = \arg\max_{c_k \in \mathcal{Y}} T_{k,i}^{(2)*}$.

25:         Partition data into $\mathcal{D}_L = \{(x_i, \hat{y}_i^{\text{OT}}) | x_i \in D, \hat{y}_i^{\text{OT}} = \hat{y}_i\}$ and $\mathcal{D}_U = \{x_i | x_i \in D, \hat{y}_i^{\text{OT}} \neq \hat{y}_i\}$.

26:         **LLPMix: LLP-Consistent Semi-Supervised Learning**

27:         **for** each mini-batch $\mathcal{B}$ from $\mathcal{D}_L, \mathcal{D}_U$ **do**

28:             For $x_u \in \mathcal{D}_U$, generate soft pseudo-label $\tilde{y}_u$ by averaging sharpened predictions from multiple weak augmentations of $x_u$.

29:             Apply MixUp to inputs and labels (both $\hat{y}_i^{\text{OT}}$ for $\mathcal{D}_L$ and guessed $\tilde{y}_u$ for $\mathcal{D}_U$) to form $\mathcal{D}_{\text{mix}}$.

30:             Compute supervised classification loss $\mathcal{L}_S$ on mixed labeled samples.

31:             Compute unsupervised consistency loss $\mathcal{L}_U$ on mixed unlabeled samples.

32:             Compute predicted bag proportions for the mini-batch (pre-MixUp) as $\hat{p}_{j,k}^{\text{batch}} = \frac{1}{|\{x_i \in \mathcal{B}_{\text{orig}} | x_i \in B_j\}|} \sum_{x_i \in \mathcal{B}_{\text{orig}}, x_i \in B_j} h_{\text{main}}(f_{\text{main}}(Aug(x_i)))_k$.

33:             Compute LLP consistency loss $\mathcal{L}_{\text{LLP-Cons}} = \frac{1}{|\mathcal{B}_{\text{bags}}|} \sum_{B_j \in \mathcal{B}_{\text{bags}}} D_{KL}(\mathbf{p}_j || \hat{\mathbf{p}}_j^{\text{batch}})$.

34:             Compute total loss $\mathcal{L}_{\text{LLPMix}} = \mathcal{L}_S + w_U \mathcal{L}_U + w_{\text{LLP}} \mathcal{L}_{\text{LLP-Cons}}$.

35:             Update $f_{\text{main}}$ and $h_{\text{main}}$ by minimizing $\mathcal{L}_{\text{LLPMix}}$.

36:         **end for**

37:     **end for**

38: **end procedure**

39: **return** $h_{\text{main}}(f_{\text{main}}(\cdot))$.

---

# D Limitations and Future Work

While RLPL demonstrates strong performance, its multi-stage framework is inherently more structured than single-stage end-to-end methods. Although our computational analysis (Section A.4) confirms that the runtime is comparable to baselines and scalable, future work could explore simplifying this refinement pipeline.

Furthermore, although RLPL is competitive in terms of average performance and robustness, other highly specialized methods might exhibit a slight advantage on specific datasets or bag sizes. Future research could also focus on extending the framework to other data modalities, such as text, or adapting it for the multi-label LLP setting.

