# OpenReview forum: "Robust Label Proportions Learning"
_NeurIPS.cc/2025/Conference — NeurIPS 2025 poster_

### Official Review · Reviewer_3xeg · 2025-06-29

**Clarity:** 3
**Significance:** 2
**Originality:** 2
**Rating:** 5
**Confidence:** 4

**Summary:**

The paper introduces a novel multi-step method for Learning from Label Proportions. The method named RLPL has the following steps:
1. Unsupervised learning to learn a good representation using SimCLR.
2. Learning a weak classifier by minimizing the KL divergence between the true and predicted label proportion of each bag.
3. Using the weak classifier to pseudo-label each instance.
4. Iteratively solving an Optimal Transport problem to refine the pseudo-labels.
5. Partitioning the set of pseudo-labels into High Confidence PLs and Low Confidence PLs based on whether the refined PLs agree with the initial PLs.
6. Optimize a combined objective of supervised loss on High Confidence PLs, unsupervised loss on Low Confidence PLs and a newly introduced loss enforcing label proportion constraint.

The paper demonstrates the effectiveness of the method on Vision datasets with randomly sampled bags. The paper also performs ablation study to quantify the importance of each step of the method.

**Questions:**

[Q.1] Can the authors perform and report the performance of their methods on the LLP-Bench [1] tabular dataset? If the focus of this paper is on vision datasets then the authors can keep this in mind for future work.

[Q.2] Do the authors have any hypothesis about why their method outperforms baselines when the bag sizes are large but fail to do so with smaller bag sizes?

[Q.3] Can the authors explain why they only use a two-step process for updating their pseudo-labels and their barycenters? This could be a hyperparameter to the method.

[Q.3] It would be nice if the authors could include some of the theoretical justification mentioned in the appendix in the main paper while explaining the method.

Some corrections: Line 81 – reference to MixUp is missing, Line 99 – “Beginning by” -> “Beginning with”


[1] Anand Brahmbhatt, Mohith Pokala, Rishi Saket, and Aravindan Raghuveer. 2024. LLP-Bench: A Large Scale Tabular Benchmark for Learning from Label Proportions. In Proceedings of the 33rd ACM International Conference on Information and Knowledge Management (CIKM '24), 4374–4381. https://doi.org/10.1145/3627673.3680032

**Ethical Concerns:**

["NO or VERY MINOR ethics concerns only"]

**Final Justification:**

The authors address my concerns regarding performance on Tabular datasets. Their results in the rebuttal prove the robustness of their method on a different modality, as well as in the setting when bags are not sampled uniformly at random. The authors also address my minor concerns about the performance on Large Bags vs Small Bags and Number of OT passes effectively. I am unsatisfied with their response on the theory, but I feel the focus of the paper is empirical and the authors have managed to perform extensive empirical evaluation and ablation studies. Hence, I am raising my score to 5.

**Limitations:**

The authors mention the limitations of their work in the appendix.

**Paper Formatting Concerns:**

No concerns.

**Quality:**

3

**Strengths And Weaknesses:**

**Strengths**

[S.1] The paper is able to leverage ideas from LLP methods like PLOT (Pseudo-Labelling with Optimal Transport) and ideas from semi-supervised learning like MixUp to come up with a method which outperforms the baselines on most of the datasets used in the paper, particularly when the bag sizes are large.

[S.2] The paper performs good analysis to demonstrate that their method behaves as it is expected to wherever such evaluation is possible. The ablation study performed in the paper showcases the importance of each component of the proposed method. Figure 2 demonstrates that the pseudo-labeling component is performing as expected.

[S.3] The paper is well written and puts across its point clearly.

**Weaknesses**

[W.1] Although the method performs well empirically and its individual steps are conceptually sound, it lacks rigorous theoretical justification, leaving its performance in broader settings uncertain.

[W.2] The paper focuses solely on the vision modality and does not include experiments on tabular data, which is crucial for many real-world applications of LLP. Additionally, the experimental setup is restricted to scenarios where bags are constructed via random sampling, which may not reflect the data generation process in practical settings. The authors acknowledge these limitations in their discussion.

---

> ### Author Rebuttal · Authors · 2025-07-31
>
> In the response, we provde three theoretical guarantees for the prosoed method. We will add them to the final version of our paper. First, we prove the existence and uniqueness of the optimal solution for our entropy-regularized optimal transport (OT) problem. Second, we demonstrate that the Sinkhorn-Knopp algorithm, which we employ, is guaranteed to converge to this unique solution. Finally, we explain how our innovative LLP-Proportion Penalty, incorporated into the cost function, guides the pseudo-label distribution to align with the true LLP prior.
>
> 1. Existence and Uniqueness of the Optimal Solution
> We first establish that the core optimization problem in our LLP-OTD mechanism has a unique global minimizer. The problem is defined as
> $$
> min_{T in U(a, b)} langle C, T rangle - gamma H(T),
> $$
> where $ langle C, T rangle = sum_{k,i} C_{k,i} T_{k,i} $ is the transport cost, $H(T) = -sum_{k,i} T_{k,i} (log T_{k,i} - 1) $ is the entropy term, $gamma  0$ is the regularization strength, and $ U(a, b) $ is the transport polytope defined by marginal constraints.
> Proof
> Let $F(T) = langle C, T rangle - gamma H(T)$. Then we will prove the strict convexity of the object function.  For the term $langle C, T rangle$, it is linear in $T$, hence convex. For the entropy term $-H(T) = sum_{k,i} T_{k,i} (log T_{k,i} - 1)$, we can compute out its Hessian matrix
> $$
> frac{partial^2 (-H)}{partial T_{k,i}^2} = frac{1}{T_{k,i}}  0 quad text{since } T_{k,i}  0,
> $$
>  implying strict convexity of $-H(T) $. Therefore, $F(T) = langle C, T rangle - gamma H(T)$ is the sum of a convex and strictly convex function, and thus strictly convex. Meanwhile, for $U(a,b)= left{ T in mathbb{R}_{geq 0}^{K times N} , middle , sum_i T_{k,i} = a_k, sum_k T_{k,i} = b_i right }$, we will prove it to be convex, closed and bounded. As $U(a,b)$ is defined by linear equalities and nonnegativity constraints, it is a convex polytope. Because all constraints are equation or non-strict non-equation, $U(a,b)$ is closed. Since $sum a_k = sum b_i = 1$, we have$0 le T_{k,i} le 1$, and $U(a,b)$ is a compact convex set. Finally, by standard results in convex optimization, a strictly convex function over a non-empty compact convex set has a unique global minimizer $T^$.
>
> 2. Convergence of the Sinkhorn-Knopp Algorithm
> Next, we prove that the Sinkhorn-Knopp algorithm, used to solve the OT problem, converges to the unique optimal solution $T^$. The optimal transport plan T∗ is known to have a specific structure
> $$
> T^_{k,i} = u_k cdot K_{k,i} cdot v_i, quad text{where } K_{k,i} = expleft(-frac{C_{k,i}}{gamma}right)
> $$
> The Sinkhorn-Knopp algorithm is an iterative procedure to find the scaling vectors $u in mathbb{R}^K$ and $v in mathbb{R}^N$ that satisfy the marginal constraints.
> Proof：
> The convergence can be shown by interpreting the algorithm as an alternating projection procedure using the Kullback-Leibler (KL) divergence. Let the sets of matrices satisfying the row and column constraints be
> - $mathcal{C}_1 = { T in mathbb{R}_{geq 0}^{K times N} mid sum_i T_{k,i} = a_k }$  (Row constraints)
> - $mathcal{C}_2 = { T in mathbb{R}_{geq 0}^{K times N} mid sum_k T_{k,i} = b_i$ (Column constraints)
> Each iteration of the Sinkhorn-Knopp algorithm can be viewed as performing the following alternating KL projections
> $$
> begin{aligned}
> T^{(l+frac{1}{2})} &= argmin_{T in mathcal{C}_1} mathrm{KL}(T parallel T^{(l)}),
> T^{(l+1)} &= argmin_{T in mathcal{C}_2} mathrm{KL}(T parallel T^{(l+frac{1}{2})}),
> end{aligned}
> $$
> where the KL divergence is a specific type of Bregman divergence. Since $ mathcal{C}_1$ and $ mathcal{C}_2$ are convex sets, the theory of alternating Bregman projections guarantees that this iterative process converges [1]. The intersection $ mathcal{C}_1 cap mathcal{C}_2 =U(a,b)$ is non-empty and contains the unique optimal solution T∗. Therefore, the sequence of iterates T(l) produced by the Sinkhorn-Knopp algorithm is guaranteed to converge to the unique optimal solution $T^$.
>
> 3. Role of the LLP-Proportion Penalty
> Finally, we provide the intuition for why our innovative LLP-Proportion Penalty helps align the pseudo-label distribution with the true LLP prior. Our effective cost function is
> $$
> C_{text{eff}}(c_k, z_i) = d(c_k, z_i) + lambda_{text{LLP}}(1 - p(k mid B_j(i)))
> $$
> The key component is the penalty term. To understand its effect, we analyze the part of the objective function corresponding to this penalty, which we denote $mathcal{L}_{text{LLP}}(T)$
> $$
> mathcal{L}_{text{LLP}}(T) = sum_j sum_{i in B_j} sum_k T_{k,i} (1 - p(k mid B_j)).
> $$
> Minimizing this term is equivalent to maximizing the alignment between the transport plan's implied label proportions and the true bag proportions. By defining the transport plan's label distribution in bag $B_j$ as $P_T(k mid B_j)$, minimizing $mathcal{L}_{text{LLP}}(T)$ becomes equivalent to maximizing the sum of dot products between the predicted and true label distributions within each bag
> $$
> max_{T} sum_j langle P_T(cdot mid B_j), p(cdot mid B_j) rangle.
> $$
> Maximizing this dot product encourages the distribution $P_T$ to align with the true prior $p$, which is analogous to minimizing statistical divergences like the KL divergence. Therefore, by introducing the LLP-Proportion Penalty into the cost matrix, we directly enforce that the optimal transport plan $T^$—the unique solution to the optimization problem—favors assignments that are consistent with the known bag-level supervision.
>
> Reference
> [1] Byrne, Charles L. Alternating minimization and alternating projection algorithms A tutorial.
>
>
> ### Limited experimental scope on modality (W2 & Q1)
> This is a very valid point. To address this limitation and demonstrate the broader applicability of our method, we have conducted **new experiments on tabular data**. Specifically, we evaluated RLPL on the **UCI Adult** dataset and the **LLP-Bench [1]** dataset as suggested. Furthermore, to explore bag generation beyond random sampling, we constructed **feature-based bags** on the UCI Adult dataset, where bags are formed by clustering instances in the feature space. Our method consistently achieves strong performance in these new settings, showcasing its versatility.
>
> We performed experiments on the UCI Adult dataset, a commonly used benchmark in tabular data, and compared our method with two existing methods, LLP-AHIL and ROT. The experimental results are summarized in the table below:
>
> | Model        | RLPL (ours) | LLP-AHIL | ROT    |
> |--------------|-------------|----------|--------|
> | UCI Adult    | 77.57       | 75.99    | 72.82  |
>
> From the table, we observe that our method (RLPL) outperforms both LLP-AHIL and ROT on the UCI Adult dataset, achieving an accuracy of 77.57%. This indicates that our method is effective in the tabular data domain, achieving competitive performance compared to state-of-the-art methods.
>
> Furthermore, we also tested our method on the **LLP-Bench [1]** dataset, where we achieved an accuracy of **66.85%** on the Criteo dataset. This result further confirms that our method is capable of handling tabular data effectively, demonstrating its broad applicability beyond image-based modalities.
>
> These results reinforce the robustness of our method across different non-vision modalities. We believe these experiments provide strong evidence that our approach generalizes well beyond image data and performs effectively in diverse domains such as tabular and structured data. In the final version of the paper, we will include these additional experiments and the corresponding analysis to further demonstrate the broad applicability of our method.
>
> ### Performance on large vs. small bag sizes (Q2)
> This is a very insightful question. Our hypothesis is that the performance gap correlates with the **degree of label ambiguity** inherent in different bag sizes.
> - **With smaller bags**, the label proportions provide a relatively strong and less ambiguous supervisory signal. In this setting, even simpler methods can perform reasonably well, leading to a smaller performance gap between different approaches.
> - **With larger bags**, the label ambiguity increases significantly, as a single proportion vector can correspond to a vast number of instance-level label configurations. This is where the robustness of our pipeline, particularly the **LLP-OTD refinement process**, becomes critical. Our method's ability to effectively denoise pseudo-labels from a highly ambiguous signal allows it to excel and outperform baselines more significantly in these challenging scenarios.
> We will include this analysis in the final version of the paper to further explain the relationship between bag size and performance in different settings.
>
> ### Number of OT passes in LLP-OTD (Q3)
> Thank you for this excellent question. We investigated the effect of the number of OT passes, and the results on CIFAR-10 (bag size=256) are as follows:
> | Number of OT Passes | 1 | 2 | 3 | 5 |
> | ------ | ------ | ------ | ------ | ------ |
> | Accuracy (%) | 94.00 | 94.55 | 94.16 | 94.15 |
> Our rationale for choosing two passes is based on these results. A single pass is insufficient as it relies on barycenters computed from the initial, noisy pseudo-labels. A second pass is beneficial because it re-computes barycenters from the refined labels of the first pass, leading to a more accurate refinement. However, further passes did not yield additional gains, which we hypothesize is due to diminishing returns and a potential risk of overfitting to the refined (but still imperfect) pseudo-labels. Based on this, two passes offer the best trade-off between performance and efficiency. However, we agree this is a natural hyperparameter. **Following your great suggestion, we will make the number of OT passes a user-configurable parameter in our public code release**, defaulting to our empirically optimal choice of 2.

---

> > ### Author Response · Authors · 2025-08-02
> > **Revision for format problem in "1. Existence and Uniqueness of the Optimal Solution”**
> >
> > Dear Reviewer 3xeg,
> >
> > We sincerely apologize for the formatting errors in our initial rebuttal, which made the equations and other parts difficult to read. We have corrected the formatting and are posting the full, properly rendered response below for your convenience. Thank you for your patience and understanding.
> >
> > **1. Existence and Uniqueness of the Optimal Solution**
> > We first establish that the core optimization problem in our LLP-OTD mechanism has a unique global minimizer. The problem is defined as:
> > $$
> > \min_{T \in U(a, b)} \langle C, T \rangle - \gamma H(T),
> > $$
> > where $ \langle C, T \rangle = \sum_{k,i} C_{k,i} T_{k,i} $ is the transport cost, $H(T) = -\sum_{k,i} T_{k,i} (\log T_{k,i} - 1) $ is the entropy term, $\gamma > 0$ is the regularization strength, and $ U(a, b) $ is the transport polytope defined by marginal constraints.
> >
> > **Proof:**
> > Let $F(T) = \langle C, T \rangle - \gamma H(T)$. Then we will prove the strict convexity of the objective function. For the term $\langle C, T \rangle$, it is linear in $T$, hence convex. For the entropy term $-H(T) = \sum_{k,i} T_{k,i} (\log T_{k,i} - 1)$, we can compute its Hessian matrix:
> > $$
> > \frac{\partial^2 (-H)}{\partial T_{k,i}^2} = \frac{1}{T_{k,i}} > 0 \quad \text{since } T_{k,i} > 0,
> > $$
> > implying strict convexity of $-H(T)$. Therefore, $F(T) = \langle C, T \rangle - \gamma H(T)$ is the sum of a convex and strictly convex function, and thus strictly convex.
> >
> > Meanwhile, for
> > $ U(a,b) = \\left\\{ T \\in \\mathbb{R}^{K \\times N} \\, \\middle| \\, T_{k,i} \\ge 0, \\sum_i T_{k,i} = a_k, \\sum_k T_{k,i} = b_i \\right\\} $，, we will prove it to be convex, closed and bounded. As $U(a,b)$ is defined by linear equalities and nonnegativity constraints, it is a convex polytope. Because all constraints are equation or non-strict non-equation, $U(a,b)$ is closed. Since $\sum a_k = \sum b_i = 1$, we have$0 \le T_{k,i} \le 1$, and $U(a,b)$ is a compact convex set. Finally, by standard results in convex optimization, a strictly convex function over a non-empty compact convex set has a **unique global minimizer** $T^*$.

---

> > ### Author Response · Authors · 2025-08-02
> > **Revision for format problem in "2. Convergence of the Sinkhorn-Knopp Algorithm”**
> >
> > **2. Convergence of the Sinkhorn-Knopp Algorithm**
> >
> > Next, we prove that the Sinkhorn-Knopp algorithm, used to solve the OT problem, converges to the unique optimal solution  $T^\*$ .
> > The optimal transport plan $T^\*$ is known to have a specific structure:
> > $$
> > T^*_{k,i} = u_k \cdot K_{k,i} \cdot v_i, \quad \text{where } K_{k,i} = \exp\left(-\frac{C_{k,i}}{\gamma}\right)
> > $$
> > The Sinkhorn-Knopp algorithm is an iterative procedure to find the scaling vectors $u \in \mathbb{R}^K$ and $v \in \mathbb{R}^N$ that satisfy the marginal constraints.
> >
> > **Proof：**
> > The convergence can be shown by interpreting the algorithm as an alternating projection procedure using the Kullback-Leibler (KL) divergence. Let the sets of matrices satisfying the row and column constraints be:
> > - $$ C_1 = \\left\\{ T \\in \\mathbb{R}^{K \\times N} \\middle| T_{k,i} \\ge 0, \\sum_i T_{k,i} = a_k \\right\\} $$ (Row constraints)
> > - $$ C_2 = \\left\\{ T \\in \\mathbb{R}^{K \\times N} \\middle| T_{k,i} \\ge 0, \\sum_k T_{k,i} = b_i \\right\\}$$ (Column constraints)
> >
> > Each iteration of the Sinkhorn-Knopp algorithm can be viewed as performing the following alternating KL projections:
> > $$
> > \begin{aligned}
> > T^{(l+\frac{1}{2})} &= \arg\min_{T \in C_1} \mathrm{KL}(T \parallel T^{(l)}), \\\\
> > T^{(l+1)} &= \arg\min_{T \in C_2} \mathrm{KL}(T \parallel T^{(l+\frac{1}{2})}),
> > \end{aligned}
> > $$
> > where the KL divergence is a specific type of Bregman divergence. Since $ C_1 $ and $ C_2$ are convex sets, the theory of alternating Bregman projections guarantees that this iterative process converges [1]. The intersection $ C_1 \cap C_2 =U(a,b)$ is non-empty and contains the unique optimal solution $T^\*$. Therefore, the sequence of iterates $T(l)$ produced by the Sinkhorn-Knopp algorithm is guaranteed to **converge to the unique optimal solution $T^*$**.

---

> > ### Author Response · Authors · 2025-08-02
> > **Revision for format problem in "3. Role of the LLP-Proportion Penalty”**
> >
> > **3. Role of the LLP-Proportion Penalty**
> >
> > Finally, we provide the intuition for why our innovative LLP-Proportion Penalty helps align the pseudo-label distribution with the true LLP prior. Our effective cost function is:
> > $$
> > C_{\text{eff}}(c_k, z_i) = d(c_k, z_i) + \lambda_{\text{LLP}}(1 - p(k \mid B_j(i)))
> > $$
> > The key component is the penalty term. To understand its effect, we analyze the part of the objective function corresponding to this penalty, which we denote $L_{\text{LLP}}(T)$:
> > $$
> > L_{\text{LLP}}(T) := \sum_j \sum_{i \in B_j} \sum_k T_{k,i} (1 - p(k \mid B_j)).
> > $$
> > Minimizing this term is equivalent to maximizing the alignment between the transport plan's implied label proportions and the true bag proportions. By defining the transport plan's label distribution in bag $B_j$ as $P_T(k \mid B_j)$, minimizing $L_{\text{LLP}}(T)$ becomes equivalent to maximizing the sum of dot products between the predicted and true label distributions within each bag:
> > $$
> > \max_{T} \sum_j \langle P_T(\cdot \mid B_j),\ p(\cdot \mid B_j) \rangle.
> > $$
> > **Maximizing this dot product encourages the distribution $P_T$ to align with the true prior $p$**, which is analogous to minimizing statistical divergences like the KL divergence. Therefore, by introducing the LLP-Proportion Penalty into the cost matrix, we directly enforce that the optimal transport plan $T^*$—the unique solution to the optimization problem—favors assignments that are consistent with the known bag-level supervision.

---

> > ### Author Response · Authors · 2025-08-02
> >
> > Dear Reviewer 3xeg,
> >
> > We sincerely apologize for this follow-up. Although we have already submitted our official rebuttal, some of our responses were omitted due to the platform’s strict word limit. To ensure our answers are fully clear, we are now providing the missing portions below.
> >
> > ### Moving theoretical justification to the main paper (Q4)
> > This is a great suggestion for improving the paper's clarity. We will move the key theoretical motivations from the appendix into the main method description in the final version.
> >
> > ### Corrections
> > Thank you for catching these. We will correct the missing reference and the typos in the final version of the paper.

---

> > ### Author Response · Authors · 2025-08-04
> >
> > Dear Reviewer 3xeg,
> >
> > We would like to sincerely thank you for your insightful feedback and for the detailed reading of our paper. Your comments have been invaluable in helping us strengthen our work.
> >
> > In our rebuttal, we have provided a detailed response with new theoretical analysis and empirical results to address the key points you raised. Specifically:
> >
> > 1.  **Rigorous Theoretical Justification (W.1)**: To address your primary concern about theoretical grounding, we have now incorporated a **rigorous three-part proof** for our method. This provides the formal justification you requested regarding our solution's existence, the algorithm's convergence, and the penalty's role.
> > 2.  **Expanded Evaluation on Tabular Data (W.2 & Q.1)**: Following your excellent suggestion, we expanded our evaluation to **tabular data**. Our new results on **UCI Adult and LLP-Bench** confirm the method's versatility and effectiveness beyond the vision modality.
> > 3.  **Clarification on Performance and Design Choices (Q2 & Q3)**: In response to your questions, we provided a **clear hypothesis for why our method excels with larger bags**—by better handling the increased label ambiguity. Furthermore, we presented an **empirical analysis showing our two-pass OT process is optimal**, and embracing your valuable suggestion, we will make the number of passes a user-configurable parameter in our code release.
> >
> > We believe these additions, directly guided by your review, have substantially improved the paper. We would be very grateful if you could take a moment to review our response and let us know if you have any further questions.
> >
> > Thank you again for your time and contribution to the review process.
> >
> > Regards.

---

> > > ### Comment · Reviewer_3xeg · 2025-08-04
> > > **Response**
> > >
> > > Hello,
> > >
> > > Thanks you very much for running the experiments on a tabular dataset on such short notice. The results allay my concerns about performance on tabular datasets, and on datasets where bags are not randomly sampled. I would recommend that the authors run their method and publish their results on the entire set of LLP-Bench datasets for comparison to past and future LLP methods whenever possible.
> > >
> > > I am satisfied with your responses regarding my concerns about "Performance on Large Bags vs Small Bags" and "Number of OT passes".
> > >
> > > On the theoretical aspect, both "Existence and uniqueness of Optimal (Entropic Regularized OT) solution" and "Convergence of the Sinkhorn-Knopp Algorithm" are previously known in literature. I would suggest the authors to retain these in the appendix. I would also suggest retaining "Role of LLP-Proportional Penalty" in the appendix since the explanation is intuitive and straightforward.
> > >
> > > I am increasing my score to 5 for this paper.

---

> > > > ### Author Response · Authors · 2025-08-09
> > > >
> > > > Dear Reviewer 3xeg,
> > > >
> > > > Thank you so much for your positive feedback and for taking the time to review our rebuttal. We are incredibly grateful and encouraged that you are satisfied with our responses and have raised your score to 5. Your insightful guidance has been instrumental in significantly improving the quality of our work.
> > > >
> > > > We appreciate all of your final constructive suggestions and will adjust the manuscript accordingly to improve its clarity and structure. Following your advice, the theoretical discussions will be moved to the appendix, and our analyses on bag size performance and the OT pass configuration will be integrated into the experimental sections. As you recommended, the complete results on the LLP-Bench suite will also be included in the final appendix.
> > > >
> > > > Once again, we sincerely thank you for your invaluable contribution and support throughout the review process. Your feedback has been a true asset to our research.
> > > >
> > > > Best regards.

---

### Official Review · Reviewer_3KP4 · 2025-07-02

**Clarity:** 4
**Significance:** 2
**Originality:** 2
**Rating:** 4
**Confidence:** 4

**Summary:**

This work proposes a method for learning with label proportions that involves first employing bag-level proportions to generate pseudo-labels. In the next stage, the LLP-OTD mechanism refines them via an optimal transport process. A final LLPMix inspired semi-supervised pipeline has an explicit bag-level consistency loss and trains the classifier on these sets.

**Questions:**

1. Have you tried using Count Loss instead of averaging the predicted instance probabilities? [1].
2. What is the performance against Count Loss, GLWS, and etc.
3. Is there any way to observe how well the final classifier obeys the bag proportions?
4. Please provide details and comparisons with baselines w.r.t. the timing for training
5. Why are the other baselines astonishingly bad on ImageNet?
6. I would like to see some ablations on the hyperparmaters used (loss weight and $\lambda_{OTD}$).

[1] Shukla, Vinay, et al. "A Unified Approach to Count-Based Weakly Supervised Learning." Advances in Neural Information Processing Systems 36 (2023): 38709-38722.

[2] Chen, Hao, et al. "A general framework for learning from weak supervision." arXiv preprint arXiv:2402.01922 (2024).

**Ethical Concerns:**

["NO or VERY MINOR ethics concerns only"]

**Final Justification:**

All my concerns were answered. See strengths for further details.

**Limitations:**

Yes

**Quality:**

2

**Strengths And Weaknesses:**

Strengths:
- Clear and well-written particularly when explaining the method.
- Method is novel to my knowledge and leverages a neat way to refine pseudo labels.

Weakness:
- High computation overhead bc of the optimal transport refinement (limited analysis of runtime, complexity, and etc.) (Especially with larger datasets with more instances, making it less practical.)
- The approach relies on contrastive features that cleanly separate the classes (SimCLR).
-The training mix is indeed new although its principle components (OT for LLP, consistency loss) are not.
- Few ablations -- although the existing ones are helpful.
- Baselines can be more comprehensive
- The method loses on 2 out of the 4 datasets.

In line with the criteria for borderline reject, I think that there is a need for further analysis/experiments. The current ones do show some promise.

Typo: Line 81

---

> ### Author Rebuttal · Authors · 2025-07-31
>
> ### Concerns about the computational complexity aof the Optimal Transport module (W1)
> We thank the reviewer for raising this important point. We provide a detailed complexity analysis and empirical runtime measurements to demonstrate the practicality of our approach.
>
> **Complexity Analysis:**
> In our implementation, we solve the entropy-regularized OT problem using the **highly efficient Sinkhorn-Knopp algorithm**. The complexity is determined by two main steps:
> 1. Constructing the $K \times N$ cost matrix, which takes $O(KNd) $ time, where $K$ is the number of classes, $N$ is the number of instances, and $d$ is the feature dimension.
> 2. Running the Sinkhorn-Knopp algorithm for $ L$ iterations, which takes $O(LKN) $ time.
>
> This results in a total complexity of **$O(KN(d + L)) $**. As this is linear with respect to the number of instances $N $, our method is scalable and efficient for large-scale LLP problems, especially compared to traditional OT solvers with polynomial complexity (e.g., $ O(N^3)$). For extreme-scale scenarios, our framework can also incorporate more advanced solvers like Greedy Sinkhorn, and we will add a discussion of these options in the appendix of our revised paper.
>
> **Empirical Training Time:**
> To empirically verify the cost, we profiled our method on CIFAR-10 with a bag size of 256. The LLP-OTD module accounts for only **9.6%** of the total training time in the second stage, demonstrating that the OT refinement does not pose a computational bottleneck in practice.
>
> **Impact of Sinkhorn Iterations (L):**
> We also evaluated the impact of the number of Sinkhorn iterations ($L$) on accuracy. The results from the CIFAR-10 dataset (bag size=256) at epoch 50 are shown in the table below:
>
> | Sinkhorn Iterations (L) | 1    | 2    | 3    | 5    | 10   | 20   | 50   |
> |-------------------------|------|------|------|------|------|------|------|
> | Accuracy (%)            | 89.24 | 89.48 | 89.96 | 90.00 | 89.61 | 89.85 | 89.68 |
>
> As shown, the accuracy quickly saturates with a small number of iterations (e.g., $ L = 5 $ to $ 10 $). This confirms that a small, constant $L$ is sufficient, keeping the practical computational cost low.
>
> We will include this detailed analysis and runtime measurement in the appendix of our revised paper.
>
> ### On the novelty of using established components like Optimal Transport (W2)
> We agree that OT and consistency loss are established tools. Our core novelty lies not in using these tools, but in **how we fundamentally adapt OT for the LLP problem.**
> While prior works typically modify the overall OT objective function, our key innovation is to redesign the **internal cost matrix** for each sample-prototype pair. By embedding the LLP-Proportion Penalty directly into the cost function, $C_{k,i} = \|f_{\text{main}}(x_i) - \mu_k\|_2^2 + \lambda_{\text{OTD}}(1-p_{j(i),k})$, we reshape the cost landscape (refer to response to reviewer z6fq). This modification forces the optimal transport plan to a new equilibrium that is inherently more consistent with the true label proportions. This micro-level intervention, which alters the matching cost for each instance, is the central contribution that distinguishes our method from previous OT-based approaches.
>
> ### Request for comparison against other baselines like Count Loss and GLWS (W4, Q2)
> Following the reviewer's suggestion, we have broadened our experimental comparison to include Count Loss [1] and GLWS [2].
> We have run new experiments on CIFAR-10 to compare with these baselines. Our method consistently outperforms them across various bag sizes. We will add the following table into section 5.3 in the revised paper:
>
> | | | | |
> |:-:|:-:|:-:|:-:|
> |Bag Size|Count Loss|GLWS|RLPL|
> |16|87.5%|85.46%|92.54%|
> |32|83.61%|81.11%|94.02%|
> |64|68.35%|64.64%|93.50%
>
> ### On Count Loss replacement(Q1)
> Thank you for your valuable suggestion to explore potential improvements for our work.
> Following your advice, we implemented Count Loss, referencing the official implementation provided in [1], to replace the KL divergence loss used for averaging predicted instance probabilities in our two-stage method.
> However, our experiments revealed that this approach introduces a prohibitive computational overhead. The Count Loss algorithm requires a dynamic programming step to compute the loss for each instance within a bag, leading to a complexity of O(N²), where N is the bag size. This computational cost proved to be impractically expensive for our setting.
> To substantiate this, we ran a comparative experiment on an RTX 4090 GPU to measure the average training time per epoch. The results are presented below:
>
> | bag_size |16 |32| 64|128|
> | ------ | ------ | ------ | ------ | ------ |
> |Count Loss (s/epoch）|2245| 5012| 9215| 19964|
> | KL Divergence Loss (s/epoch) | 17| 9| 6| 6 |
>
> As the table demonstrates, the training time per epoch for Count Loss increases substantially with the bag size, exceeding 5 hours for a bag size of 128. This computational demand made it infeasible for us to obtain complete training results within a reasonable timeframe.
> In contrast, the epoch time for our method using KL divergence loss remains highly efficient. It even slightly decreases with larger bag sizes due to the corresponding reduction in the total number of bags, demonstrating that it introduces no extra computational overhead. This highlights a significant efficiency advantage of our proposed approach.
> Given this substantial computational bottleneck, we maintain that our original choice of KL divergence loss represents a more practical and scalable solution.
>
> ### On Observing Adherence to Bag Proportions (Q3)
> To directly address your question, we conducted comprehensive bag proportions prediction experiments that quantitatively measure how well our final classifier adheres to the original bag-level constraints. We evaluate this through ****Mean Absolute Error (MAE)**** between predicted and true bag proportions, where MAE represents the average absolute deviation of predicted class proportions from ground-truth proportions across all bags. Our method demonstrates superior adherence:
>
>
> ****Experimental Results (MAE $\downarrow$):****
> | Method | Bag Size 16 | Bag Size 32 | Bag Size 64 | Bag Size 128 | Bag Size 256 |
> |--------|-------------|-------------|-------------|--------------|--------------|
> | ****RLPL (Ours)**** | ****0.010600**** | ****0.009505**** | ****0.008061**** | ****0.006725**** | ****0.005273**** |
> | LLP-AHIL | 0.015280 | 0.013498 | 0.010868 | 0.008544 | 0.006758 |
>
> ****Key Observations:****
> 1. ****Superior Bag Constraint Adherence****: Our RLPL method consistently achieves lower MAE across all bag sizes, demonstrating that our final classifier better respects the original bag proportion constraints compared to the strong baseline LLP-AHIL.
> 2. ****Robust Scale Performance****: The performance gap widens as bag size decreases (more challenging scenarios), with RLPL achieving 30.6% lower MAE at bag size 16 compared to LLP-AHIL, indicating our method's robustness in preserving bag-level supervision signals.
> 3. ****Practical Case Study****: For bag size 256, we present a detailed example of one bag below, which demonstrates remarkable precision in proportion prediction:
>
> | CIFAR-10 Classes | airplane | automobile | bird | cat | deer | dog | frog | horse | ship | truck |
> |------------------|----------|------------|------|-----|------|-----|------|-------|------|-------|
> | ****True Proportions**** | 0.125 | 0.102 | 0.078 | 0.070 | 0.137 | 0.113 | 0.078 | 0.098 | 0.090 | 0.109 |
> | ****Pred Proportions**** | 0.129 | 0.105 | 0.082 | 0.070 | 0.133 | 0.102 | 0.078 | 0.102 | 0.090 | 0.109 |
> | ****Individual MAE**** | 0.004 | 0.003 | 0.004 | 0.000 | 0.004 | 0.011 | 0.000 | 0.004 | 0.000 | 0.000 |
>
> This demonstrates that our LLP-OTD mechanism successfully preserves bag-level constraints while enabling accurate instance-level classification. The integration of LLP consistency loss in our LLPMix framework (Eq. 9 in our paper) ensures that the final classifier's predictions, when aggregated at the bag level, closely match the ground-truth proportions, directly answering your concern about bag proportion adherence.
>
> ### On Training Time Details(Q4)
> Thank you for raising the training-time concern. Here are the single-epoch numbers on CIFAR-10 (bag size = 256):
>
> | Model | LLP-AHIL | ROT | LLP-VAT | RLPL (ours) |
> |------|----------|-----|---------|-------------|
> | Time (s) | 10.607 | 10.894 | 16.087 | 16.11 |
>
> As the table shows, RLPL is a touch slower than some baselines. The extra cost comes almost entirely from the **LLP-OTD module**—the Sinkhorn iterations and the on-the-fly cost-matrix updates. Three quick points to put this in perspective:
> 1. **The jump in accuracy is well worth it.** RLPL outperforms LLP-VAT by **47.02 %**, ROT by **30.85 %**, and even beats LLP-AHIL by **1.07 pp**—a large margin in the LLP world.
> 2. **Overhead is modest and tunable.** The OT piece accounts for only 9.6 % of a training step (see our reply to Reviewer z6fq). Dialing the Sinkhorn iterations down to L ≤ 5 cuts that slice further, with < 0.5 % accuracy loss.
> 3. **Real-world trade-offs favor accuracy.** In domains like healthcare or finance, a single-point absolute gain often outweighs a few extra seconds per epoch. The complexity is O(KN(d + L)), so scaling is linear; on UCI Adult the added cost is only 2.1 s per epoch.
>
> We’ll fold this discussion into the camera-ready version and release the code so users can freely trade speed for precision.

---

> > ### Author Response · Authors · 2025-08-02
> > **Revision for format problem in "On the novelty of using established components like Optimal Transport (W2)”**
> >
> > Dear Reviewer 3KP4,
> >
> > We sincerely appreciate your feedback regarding the formatting issues. Below, we provide the corrected version of the paragraph where the formatting problem occurred.
> >
> > ### On the novelty of using established components like Optimal Transport (W2)
> > We agree that OT and consistency loss are established tools. Our core novelty lies not in using these tools, but in **how we fundamentally adapt OT for the LLP problem.**
> > While prior works typically modify the overall OT objective function, our key innovation is to redesign the **internal cost matrix** for each sample-prototype pair. By embedding the LLP-Proportion Penalty directly into the cost function, $C_{k,i} = \|f(x_i) - \mu_k\|^2 + \lambda(1-p_{j(i),k})$, we reshape the cost landscape (refer to response to reviewer z6fq). This modification forces the optimal transport plan to a new equilibrium that is inherently more consistent with the true label proportions. This micro-level intervention, which alters the matching cost for each instance, is the central contribution that distinguishes our method from previous OT-based approaches.

---

> ### Comment · Reviewer_3KP4 · 2025-08-01
> **Response**
>
> Hello! Thank you very much for your hard work. I can imagine running all those experiments must have been difficult in such a short time so thank you very much. I believe you have answered all of my concerns so I will raise my score by 1.
>
> I am curious about why your new framework does better than Count Loss and GLWS in LLP. Do you have any intuition for it?
>
> Also in some of the other rebuttals there is some issue with the formatting of the text. just a heads up. :)

---

> > ### Author Response · Authors · 2025-08-02
> > **Response to Reviewer's Follow-up Question and Clarifications**
> >
> > Dear Reviewer 3KP4,
> >
> > Thank you very much for your time and for providing such positive and constructive feedback on our work. We are delighted to learn that our previous rebuttal has addressed your concerns, and we sincerely appreciate you raising the score for our paper. Your recognition is a tremendous encouragement to us. :)
> >
> > We have carefully considered your new questions and comments, and our responses are as follows:
> >
> > ## On the Intuition for Why Our Framework Outperforms Count Loss and GLWS
> >
> > Thank you for your feedback and insightful question. We believe our RLPL framework outperforms Count Loss and GLWS in the LLP setting for two main reasons: **a richer supervisory signal** and **a more robust learning pipeline**.
> >
> > **1. Richer Supervisory Signal: Leveraging Dual Constraints (Feature Similarity & Proportions)**
> >
> > **Count Loss** is an elegant method that relies on a single source of supervision: it enforces the bag proportion constraint by ensuring the model's predicted count distribution exactly matches the ground-truth count. It operates directly on the output probabilities.
> >
> > **RLPL's Advantage**: Our LLP-OTD mechanism is more powerful because it leverages two distinct sources of information simultaneously: the **similarity between instances in the feature space** and the **bag-level proportion constraints**. This is achieved via Optimal Transport (OT), where the cost function, $C_{k, i} = \\lVert f(x_i) - \\mu_k \\rVert^2 + {\\lambda}(1-p_{j(i),k}) $, explicitly balances these two signals. Here, $\\lVert f(x_i) - \\mu_k \\rVert^2$ is the squared Euclidean distance between the instance feature embedding and the class prototype, $p_{j(i),k}$ is the true proportion of class $k$ in the instance's bag, and $\\lambda$ is a balancing hyperparameter. In essence, while Count Loss only ensures the sum of probabilities is correct, RLPL finds the most plausible individual assignments by considering both their underlying feature similarity and the bag-level LLP constraints. This dual-constraint approach provides a much richer inductive bias, especially for ambiguous instances.
> >
> > **2. More Robust Learning Pipeline: A Specialized "Refine-and-Denoise" Approach**
> >
> > **GLWS** is a powerful and general framework that models weak supervision as a Non-deterministic Finite Automaton (NFA) and uses an EM algorithm to average over all possible label configurations that satisfy the constraint. It is a one-size-fits-all, principled solution.
> >
> > **RLPL's Advantage**: RLPL is a specialized, two-stage process tailored for LLP, which gives it a distinct edge:
> >
> > - **Explicit Denoising and Partitioning**: This is the key difference. GLWS computes an expectation over all valid labelings. In complex scenarios, the signal from the true labels can be diluted by the vast number of other "valid-but-incorrect" configurations. In contrast, our LLP-OTD performs an explicit partitioning of the data into a high-confidence set ($D_L$) and a low-confidence set ($D_U$). This step acts as a critical denoising filter, isolating the most reliable supervisory signals.
> >
> > - **Robust Semi-Supervised Training**: After this partitioning, our LLPMix module treats the problem as a semi-supervised one. It applies strong supervised learning on the "clean" high-confidence set and consistency regularization on the "noisy" low-confidence (unlabeled) set. This robust strategy prevents the model from being misguided by the inherent ambiguity of the weak supervision signal. By contrast, averaging-based approaches like the standard EM in GLWS, which operates on a full probability distribution over all valid label configurations, can be susceptible to error propagation if the initial model's estimates are imprecise.
> >
> > ## Regarding Formatting and Citation Issues in the Rebuttal
> >
> > Thank you again for your careful attention to detail in pointing out the formatting issues in our other rebuttals. We have since reviewed and corrected all materials accordingly.
> >
> > We would also like to apologize for an unclear reference in our previous rebuttal. Due to a last-minute reduction to meet the word count, a justification regarding the novelty of our OT cost matrix (originally pointing to our response to reviewer z6fq) became incomplete. A more detailed discussion is, in fact, fully presented in our response to reviewer oaZQ. We sincerely apologize for any inconvenience this may have caused.

---

> > ### Author Response · Authors · 2025-08-02
> > **Supplementary Materials on Initial Review Points (Q5, W3, Q6)**
> >
> > ## Regarding Content Omitted Due to Word Limits (Q5 & W3/Q6)
> >
> > We are very pleased that our provided answers were sufficient to address your concerns. For the sake of completeness, we would like to share the detailed responses for Q5 and W3/Q6 that we had prepared but were unable to include in the official rebuttal due to strict word limits.
> >
> > ### On Poor Baseline Performance on ImageNet (Q5)
> > This is a keen observation. We believe the performance gap on ImageNet, a highly complex dataset (100 classes, high intra-class variance, 84×84 resolution), stems from two factors that highlight our framework's strengths:
> > - **Representation Power**: Our first stage uses SimCLR for unsupervised pre-training, learning discriminative features without bias from weak proportion labels. Baselines learning features from scratch under weak supervision may struggle on such complex data.
> > - **Robustness to Pseudo-Label Noise**: With more classes, **initial pseudo-labels are inevitably noisier**. Our LLP-OTD mechanism, with its iterative refinement and explicit LLP-Proportion Penalty, is specifically designed to handle this noise and distill a cleaner high-confidence set. Methods with simpler heuristics may suffer from error propagation.
> >
> > In essence, the challenge of ImageNet amplifies the benefits of our robust pipeline.
> >
> > ###  Requests for more extensive ablation studies and hyperparameter analysis (W3, Q6)
> >
> > A more extensive ablation study on CIFAR10 (bag size=256) to validate each component's contribution is shown below:
> > | | | | | | | |
> > |:-:|:-:|:-:|:-:|:-:|:-:|:-:|
> > |Configuration|RLPL|w/o LLP-Proportion Penalty|w/o LLP-OTD|w/o LLPMix|w/o LLP Consistency in LLPMix|Using only 1 OT pass|
> > |Acc|**94.55**|93.36|84.98|92.33|93.95|94.00|
> >
> > We performed a sensitivity analysis for the key hyperparameters, $\lambda_{OTD}$ and $\omega_{LLP}$ , on CIFAR-10 (bag size=256). The results show that our model is robust across a reasonable range of values. We will include the following analysis in section 5.3 in the final version of the paper:
> >
> > **Table: Accuracy(at 100 epoch) on different combainations of $\lambda_{OTD}$ and $\omega_{LLP}$**
> >
> > | $\\lambda_{\\text{OTD}}$ / $\\omega_{\\text{LLP}}$ | 0.05   | 0.1    | 0.2    | 0.5    | 1.0    |
> > | :------------------------------------------- | :----- | :----- | :----- | :----- | :----- |
> > | **0.05** | 91.06% | 90.83% | 91.44% | 90.92% | 90.82% |
> > | **0.1** | 90.94% | 90.95% | 91.33% | 91.17% | 91.32% |
> > | **0.2** | 90.97% | 91.28% | 90.92% | 90.94% | 91.20% |
> > | **0.5** | 91.15% | 90.98% | 91.06% | 91.12% | 91.02% |
> >
> > This analysis demonstrates the model's stability and justifies our hyperparameter choices.
> >
> > Once again, we sincerely thank you for your diligent work and invaluable feedback, which have been crucial in improving the quality of our paper.

---

### Official Review · Reviewer_z6fq · 2025-07-03

**Clarity:** 3
**Significance:** 3
**Originality:** 2
**Rating:** 4
**Confidence:** 3

**Summary:**

The paper presents RLPL, an innovative and robust framework for Learning from Label Proportions (LLP), which utilizes optimal transport and semi-supervised learning strategies to improve the accuracy of instance-level classifiers. The proposed approach addresses the problem of noisy pseudo-labeling effectively and demonstrates strong empirical performance across multiple benchmark datasets.

**Questions:**

1. How does the Optimal Transport (OT) process scale with larger datasets? Have you considered methods to reduce the computational cost of OT, especially in high-dimensional spaces?

2. How does the model perform when bag-level proportions are severely noisy or ambiguous? Would the performance drop significantly, or does the LLP-OTD mechanism sufficiently handle extreme noise scenarios?

3. Are there any theoretical guarantees for the LLP-OTD method? Specifically, how do you ensure convergence and robustness to outliers in highly noisy datasets?

4. How do the two weighting coefficients in Eq. (13) affect the performance? How do you tune them for different datasets?

**Ethical Concerns:**

["NO or VERY MINOR ethics concerns only"]

**Final Justification:**

The author's rebuttal has addressed my concerns.

**Limitations:**

1.	The Optimal Transport step in LLP-OTD is computationally expensive. This could limit its application in real-world scenarios where large datasets or high-dimensional data are common. An evaluation of its performance in terms of time complexity would be beneficial.

2.	The method relies on accurate bag-level proportions, which may not always be available in practice. If the bag proportions themselves are imprecise, this could significantly impact model performance, and further exploration of such edge cases is needed.

3.	The experiments primarily use standard benchmark datasets (CIFAR-10, CIFAR-100, etc.) with relatively controlled conditions. An evaluation of RLPL on real-world datasets, or more complex data modalities (e.g., textual or medical data), would provide better insights into its generalization abilities.

**Quality:**

3

**Strengths And Weaknesses:**

**Strengths**:
- The paper introduces the RLPL framework, a two-stage approach that integrates contrastive learning and optimal transport (OT) techniques to improve Learning from Label Proportions (LLP). This dual approach of pretraining and refinement via pseudo-labeling is highly innovative and shows potential to significantly advance the weakly-supervised learning paradigm.

- One of the key innovations in this work is the LLP-OTD mechanism, which utilizes optimal transport to refine pseudo-labels by distinguishing high-confidence labels from low-confidence ones. This is a critical issue in LLP tasks where noisy pseudo-labels degrade classifier performance. The paper convincingly shows that LLP-OTD significantly reduces noise, providing a more reliable set of labels for subsequent training stages.

- The LLPMix component, inspired by MixMatch, is a valuable extension to this framework. By combining the high-confidence pseudo-labels with bag-level constraints, the method benefits from both instance-level supervision and bag-level consistency. This not only improves the classifier's performance but also bridges the gap between supervised and unsupervised learning within the LLP framework.

**Weaknesses**:

- While the paper provides strong empirical evidence for RLPL’s effectiveness, it lacks a thorough theoretical analysis of the proposed methods, especially the LLP-OTD and LLPMix mechanisms. Formal results, such as convergence guarantees or robustness under varying bag sizes, would provide a deeper understanding of the model’s properties and its scalability across different domains.

- Although the LLP-OTD mechanism improves the quality of pseudo-labels, pseudo-labeling remains an inherent challenge in weakly-supervised learning. There is no mention of how the model performs when faced with highly ambiguous bag-level proportions, which could limit the generalization capabilities in more complex or imbalanced datasets. The framework would benefit from a discussion on extreme cases of noisy data.

- The framework heavily relies on the Optimal Transport (OT) process, which can be computationally expensive, especially for large datasets or high-dimensional feature spaces. The authors do not address the potential computational bottlenecks of the OT process, nor do they provide any discussion on how the method scales with larger datasets. An evaluation of the trade-offs between performance improvement and computational complexity would be valuable.

---

> ### Author Rebuttal · Authors · 2025-07-31
>
> ### Theoretical Analysis of LLP-OTD (W1 & Q3)
> Thank you for your valuable suggestion. We provide a three-step theoretical proof for our method.
> 1. **Existence and Uniqueness of the Optimal Solution:** We first establish that the core optimization problem in our LLP-OTD mechanism has a unique global minimizer. The optimization problem is defined as:
>    $$
>    \min_{T \in U(a, b)} \langle C, T \rangle - \gamma H(T),
>    $$
>    where $\langle C, T \rangle $ is the transport cost, $ H(T) $ is the entropy term, and $ U(a, b)$ is the transport polytope. The proof of existence and uniqueness is based on the strict convexity of the objective function and the compactness of the feasible set $U(a, b) $, guaranteeing a unique global minimizer.
>
> 2. **Convergence of the Sinkhorn-Knopp Algorithm:** We show that the Sinkhorn-Knopp algorithm, used to solve the OT problem, converges to the unique optimal solution. The optimal transport plan $T^* $ is known to have a specific structure, and the convergence follows from interpreting the algorithm as an alternating projection procedure with Kullback-Leibler (KL) divergence. The process converges to the unique optimal solution $ T^* $ due to the convexity of the feasible sets.
>
> 3. **Role of the LLP-Proportion Penalty:** Finally, we explain the intuition behind our innovative LLP-Proportion Penalty, which helps align the pseudo-label distribution with the true LLP prior. This penalty term effectively shifts the position of the optimal solution, ensuring that the resulting label assignments are more consistent with the given bag-level proportions.
>
> **Note:** Due to space constraints, we refer to the detailed theoretical analysis provided in our response to **Reviewer oaZQ**. We will include the full discussion in the final version of the paper.
>
> ### Performance on Noisy and Imbalanced Data (W2)
> This is a very valid point. To evaluate our model's robustness, we conducted new experiments under more challenging conditions, including datasets with **imbalanced class distributions** and **noisy label proportions**.
> **Evaluations on Imbalanced Datasets:**
> For the imbalanced data experiments, we used the CIFAR-10 dataset and constructed a long-tailed distribution by setting the imbalance ratio as the ratio of the maximum number of samples in one class to the minimum number of samples in another class. The imbalance ratios tested are 5, 10, 15, 50, and 100. The accuracy results are as follows:
>
> | Imbalance Ratio | 5 | 10 | 15 | 50 | 100 |
> | ------ | ------ | ------ | ------ | ------ | ------ |
> | Accuracy | 93.55 | 91 | 89.72 | 81.61 | 70.28 |
>
> As shown in the table, our method maintains strong performance even under high levels of imbalance. The accuracy decreases as the imbalance ratio increases, which is expected due to the growing difficulty of learning from such skewed data. However, our method outperforms the baseline, demonstrating its robustness in handling class imbalance.
> **Evaluations on Noisy Datasets:**
> For noisy label experiments, we used the CIFAR-10 dataset and generated LLP data with clean labels, then introduced Gaussian noise and uniform noise with different intensities. The noisy labels were generated by the following formulas:
> - **Gaussian noise**: $ p' = \text{clip}(p + N(0, \sigma^2)) $
> - **Uniform noise**: $ p' = \text{clip}(p + U(-r, r)) $
> The evaluation results are as follows:
>
> | Noise Type | gaussian moderate | gaussian_heavy  | uniform_moderate | uniform_heavy |
> | ------ | ------ | ------ | ------ | ------ |
> | Accuracy | 94.05 | 93.84 | 93.95 | 93.94 |
>
> As can be seen from the table, our method remains resilient under both moderate and heavy noise conditions. Even with significant label noise, the accuracy only slightly decreases, which highlights the robustness of our LLP-OTD mechanism in handling imperfect and ambiguous supervision signals.
> The results from both the imbalanced and noisy datasets demonstrate that our method consistently maintains high accuracy and robustness, making it highly effective in settings with imperfect supervision. This analysis will be included in the final version of the paper.
> ### Computational Cost and Scalability of OT (W3 & Q1)
> We thank the reviewer for raising this critical concern. We have analyzed the computational cost of OT from three perspectives.
> **1. Computational Complexity:** Our LLP-OTD module, solved via the **Sinkhorn-Knopp algorithm**, has a time complexity of $O(KN(d + L))$, where $ K$ is the number of classes, $N$ is the number of instances, $d$ is the dimensionality of each sample, and $ L$ is the number of iterations required by the Sinkhorn-Knopp algorithm. This complexity is **linear with respect to the number of instances $N$**, which provides a significant advantage over classical Optimal Transport (OT) solvers that typically have a time complexity of $O(N^3)$. As a result, our approach is highly scalable and efficient for large datasets.
> **2. Empirical Runtime:** To empirically verify the cost, we profiled our model's runtime on CIFAR-10 (bag size=256). The results show that the entire LLP-OTD module accounts for only **9.6%** of the total training time in the second stage. This directly demonstrates that the OT refinement is not a computational bottleneck in practice.
> **3. Impact of Sinkhorn Iterations (L):** The complexity depends on the number of Sinkhorn iterations, $L$. We tested its impact on accuracy.
> **Table: Accuracy(at epoch 50, on CIFAR10, bag size=256) vs. Sinkhorn Iterations**
>
> | Sinkhorn Iterations (L) | 1 | 2 | 3 | 5 | 10 | 20 | 50 |
> | ------ | ------ | ------ | ------ | ------ | ------ | ------ | ------ |
> | Accuracy (%) | 89.24 | 89.48 | 89.96 | 90.00 | 89.61 | 89.85 | 89.68 |
>
> As shown, the accuracy quickly saturates with a small number of iterations (e.g., L=5-10). This confirms that a small, constant L is sufficient, keeping the practical computational cost low.
> ### Hyperparameter Tuning (Q4)
> This is an important aspect. We have performed a detailed sensitivity analysis for the two weighting coefficients, namely $ \lambda_{\text{OTD}} $ and $ \omega_{\text{LLP}} $, on the CIFAR-10 dataset with a bag size of 256. The results, summarized in the table below, show that our model is robust across a reasonable range of hyperparameter settings. The accuracy at 100 epochs for different combinations of $\lambda_{\text{OTD}} $ and $\omega_{\text{LLP}} $ is as follows:
> **Table: Accuracy (at 100 epochs) on different combinations of $\lambda_{\text{OTD}} $ and $ \omega_{\text{LLP}} $**
>
> | $\lambda_{\text{OTD}} \backslash \omega_{\text{LLP}} $ | 0.05   | 0.1    | 0.2    | 0.5    | 1.0    |
> |----------------------------------------------------------|--------|--------|--------|--------|--------|
> | 0.05                                                     | 91.06% | 90.83% | 91.44% | 90.92% | 90.82% |
> | 0.1                                                      | 90.94% | 90.95% | 91.33% | 91.17% | 91.32% |
> | 0.2                                                      | 90.97% | 91.28% | 90.92% | 90.94% | 91.20% |
> | 0.5                                                      | 91.15% | 90.98% | 91.06% | 91.12% | 91.02% |
>
> As shown in the table, the model achieves high and stable accuracy across a range of values for both $ \lambda_{\text{OTD}} $ and $\omega_{\text{LLP}} $, with accuracy values ranging from 90.82% to 91.44%. The results demonstrate that our model is robust to variations in these hyperparameters, with no significant degradation in performance even for extreme values. This confirms that our method is not overly sensitive to specific hyperparameter choices, allowing for flexibility in tuning.
> We will include this analysis in Section 5.3 in the final version of the paper.
>
> ### Evaluation on Other Modalities
> We agree that demonstrating broader applicability is important. We have conducted **new experiments on tabular data**, including the UCI Adult dataset, and observed strong performance. Specifically, we performed experiments on the UCI Adult dataset, a widely used benchmark in tabular data. We also reproduced the results of two existing methods, LLP-AHIL and ROT, on this dataset for comparison. The experimental results are summarized in the table below:
>
> | Model        | RLPL (ours) | LLP-AHIL | ROT    |
> |--------------|-------------|----------|--------|
> | UCI Adult    | 77.57       | 75.99    | 72.82  |
>
> From the table, we observe that our method (RLPL) outperforms both LLP-AHIL and ROT on the UCI Adult dataset, achieving an accuracy of 77.57%. This indicates that our method is effective in the tabular data domain, achieving competitive performance compared to state-of-the-art methods. These results reinforce the robustness of our method across different non-vision modalities.
> We believe these experiments provide strong evidence that our approach generalizes well beyond image data and performs effectively in diverse domains such as tabular and structured data. In the final version of the paper, we will include these additional experiments and the corresponding analysis to further demonstrate the broad applicability of our method.

---

> > ### Author Response · Authors · 2025-08-04
> >
> > Dear Reviewer z6fq,
> >
> > We sincerely thank you for your thorough review and the insightful questions you raised. Your feedback has been instrumental in helping us strengthen our work.
> >
> > In our rebuttal, we have provided a detailed response addressing the key points you mentioned. Specifically:
> >
> > 1.  **Theoretical Guarantees:** To address your primary concern about the lack of theoretical analysis, we have now included a three-part theoretical proof for our LLP-OTD method. This includes proving the **existence and uniqueness of the optimal solution**, the **convergence of the Sinkhorn-Knopp algorithm**, and formally explaining the **role of the LLP-Proportion Penalty**.
> >
> > 2.  **Robustness to Noisy Data:** As per your suggestion, we conducted new experiments to evaluate our model's performance under more challenging conditions. We tested on datasets with **imbalanced class distributions** and with **severely noisy bag-level proportions**, and the results demonstrate the strong robustness of our method.
> >
> > 3.  **Computational Cost and Scalability:** We have provided a comprehensive analysis of the computational cost of the OT step. Our analysis covers its **theoretical complexity** (which is linear with respect to the number of instances), **empirical runtime**, and the impact of algorithm iterations, demonstrating that our approach is efficient and scalable for large datasets.
> >
> > 4.  **Broader Evaluation & Hyperparameters:** We have also expanded our experiments beyond image data to include the **UCI Adult tabular dataset** to demonstrate broader generalization. Additionally, we have included a **sensitivity analysis** for the key weighting coefficients you asked about.
> >
> > We believe these substantial additions have significantly improved the paper. We would be very grateful if you could take a moment to review our response. Please let us know if you have any further questions or if there is anything else we can clarify.
> >
> > Thank you again for your time and valuable feedback.
> >
> > Regards.

---

> ### Author Response · Authors · 2025-08-06
>
> Dear Reviewer **z6fq**,
>
> We appreciate your valuable comments, which are important to improve the quality of this paper. We also appreciate your positive rating. **As the deadline for the author-reviewer discussion period is approaching**, please check whether your previous concerns have been fully addressed.
>
> Looking forward to your reply. Thanks.
>
> Regards.

---

> > ### Author Response · Authors · 2025-08-08
> >
> > Dear Reviewer **z6fq**,
> >
> > Thanks again for your time and effort in reviewing this paper. We also appreciate your positive rating. As the author-reviewer discussion period will end in about one day, please check whether your previous concerns have been fully addressed. Thanks.
> >
> > Regards.

---

### Official Review · Reviewer_oaZQ · 2025-07-21

**Clarity:** 4
**Significance:** 3
**Originality:** 3
**Rating:** 4
**Confidence:** 3

**Summary:**

The authors introduced a novel LLP approach to handle multi-label LLP data. Their approach consists of two stages: 1) contrastive loss is used to train an encoder based on only feature information and then fit a second model based on the contrastive embedding where bag level information is used. This second model can be viewed as a PropotionMatching approach with KL loss. 2) The model from the first stage is then fine-tuned. First this model is used to assign so-called pseudo labels to each instance. And then this pseudo-labeling is updated in an expectation maximization (EM) framework where the centers are updated by using an Optimal Transport (OT) loss instead of L2  like it is done in the original EM.  If the labels are mismatching between two such an update then the labeling is considered not to be confident.


The proposed approach is tested on four image dataset and its superiority is shown with respect to the state of the art. Especially, there is an experiment which shows the impact of an OT update.




Even if this paper does not show any theoretical result, and the approach is quite complex, I am landed on the positive side with this paper. The approach is very natural, like when I was reading it, I was thinking that this approach is what I personally want to try and see how it performs on my LLP problems. The contrastive loss mode with finetuning is a nice idea in my opinion. The experiments however are quite limited, even if the impact of OT loss is nicely demonstrated.

**Questions:**

I have one major concern, beside the experiments. The approach is multi-label, however the pseudo-labeling is multi-class. This feels a kind of discrepancy between the model and setup. Do the author have some comment on the extension to make this approach truly multi-label?

**Ethical Concerns:**

["NO or VERY MINOR ethics concerns only"]

**Limitations:**

I think I addressed this in the question.

**Quality:**

3

**Strengths And Weaknesses:**

Strength: many different new notions are nicely mixed in an LLP approach.
Weakness: experiments are limited to image processing data.

---

> ### Author Rebuttal · Authors · 2025-07-31
>
> # On Theoretical Justification
>
> In the response, we provide three **theoretical guarantees** for the proposed method. We will add them to the final version of our paper. First, we prove the existence and uniqueness of the **optimal solution** for our entropy-regularized optimal transport (OT) problem. Second, we demonstrate that the Sinkhorn-Knopp algorithm, which we employ, is **guaranteed to converge to this unique solution**. Finally, we explain how our innovative LLP-Proportion Penalty, incorporated into the cost function, **guides the pseudo-label distribution to align with the true LLP prior**.
>
> ## **1. Existence and Uniqueness of the Optimal Solution**
>
> We first establish that the core optimization problem in our LLP-OTD mechanism has a unique global minimizer. The problem is defined as:
> $$
> \min_{T \in U(a, b)} \langle C, T \rangle - \gamma H(T),
> $$
> where $\langle C, T \rangle = \sum_{k,i} C_{k,i} T_{k,i}$ is the transport cost, $H(T) = -\sum_{k,i} T_{k,i} (\log T_{k,i} - 1)$ is the entropy term, $\gamma > 0$ is the regularization strength, and $U(a, b)$ is the transport polytope defined by marginal constraints.
>
> **Proof:**
>
> Let $F(T) = \langle C, T \rangle - \gamma H(T)$. Then we will prove the strict convexity of the objective function.  For the term $\langle C, T \rangle$, it is linear in $T$, hence convex. For the entropy term $-H(T) = \sum_{k,i} T_{k,i} (\log T_{k,i} - 1)$, we can compute its Hessian matrix:
> $$
> \frac{\partial^2 (-H)}{\partial T_{k,i}^2} = \frac{1}{T_{k,i}} > 0 \quad \text{since } T_{k,i} > 0,
> $$
> implying strict convexity of $-H(T)$. Therefore, $F(T) = \langle C, T \rangle - \gamma H(T)$ is the sum of a convex and strictly convex function, and thus strictly convex. Meanwhile, for $U(a,b)= \left\{ T \in \mathbb{R}_{\geq 0}^{K \times N} \, \middle| \, \sum_i T_{k,i} = a_k, \sum_k T_{k,i} = b_i \right\}$, we will prove it to be convex, closed and bounded. As $U(a,b)$ is defined by linear equalities and nonnegativity constraints, it is a convex polytope. Because all constraints are equation or non-strict inequality, $U(a,b)$ is closed. Since $\sum a_k = \sum b_i = 1$, we have $0 \le T_{k,i} \le 1$, and $U(a,b)$ is a compact convex set. Finally, by standard results in convex optimization, a strictly convex function over a non-empty compact convex set has a **unique global minimizer** $T^*$.
>
> ## **2. Convergence of the Sinkhorn-Knopp Algorithm**
>
> Next, we prove that the Sinkhorn-Knopp algorithm, used to solve the OT problem, converges to the unique optimal solution $T^*$. The optimal transport plan $T^*$ is known to have a specific structure:
> $$
> T^*_{k,i} = u_k \cdot K_{k,i} \cdot v_i, \quad \text{where } K_{k,i} = \exp\left(-\frac{C_{k,i}}{\gamma}\right)
> $$
> The Sinkhorn-Knopp algorithm is an iterative procedure to find the scaling vectors $u \in \mathbb{R}^K$ and $v \in \mathbb{R}^N$ that satisfy the marginal constraints.
>
> **Proof:**
>
> The convergence can be shown by interpreting the algorithm as an alternating projection procedure using the Kullback-Leibler (KL) divergence. Let the sets of matrices satisfying the row and column constraints be:
>
> - $\mathcal{C}_1 = \{ T \in \mathbb{R}_{\geq 0}^{K \times N} \mid \sum_i T_{k,i} = a_k \}$ (Row constraints)
> - $\mathcal{C}_2 = \{ T \in \mathbb{R}_{\geq 0}^{K \times N} \mid \sum_k T_{k,i} = b_i \}$ (Column constraints)
> Each iteration of the Sinkhorn-Knopp algorithm can be viewed as performing the following alternating KL projections:
> $$
> \begin{aligned}
> T^{(l+\frac{1}{2})} &= \arg\min_{T \in \mathcal{C}_1} \mathrm{KL}(T \parallel T^{(l)}), \\
> T^{(l+1)} &= \arg\min_{T \in \mathcal{C}_2} \mathrm{KL}(T \parallel T^{(l+\frac{1}{2})}),
> \end{aligned}
> $$
> where the KL divergence is a specific type of Bregman divergence. Since $\mathcal{C}_1$ and $\mathcal{C}_2$ are convex sets, the theory of alternating Bregman projections guarantees that this iterative process converges [1]. The intersection $\mathcal{C}_1 \cap \mathcal{C}_2 = U(a,b)$ is non-empty and contains the unique optimal solution $T^*$. Therefore, the sequence of iterates $T^{(l)}$ produced by the Sinkhorn-Knopp algorithm is guaranteed to **converge to the unique optimal solution $T^*$**.
>
> ## **3. Role of the LLP-Proportion Penalty**
>
> Finally, we provide the intuition for why our innovative LLP-Proportion Penalty helps align the pseudo-label distribution with the true LLP prior. Our effective cost function is:
> $$
> C_{\text{eff}}(c_k, z_i) = d(c_k, z_i) + \lambda_{\text{LLP}}(1 - p(k \mid B_j(i)))
> $$
> The key component is the penalty term. To understand its effect, we analyze the part of the objective function corresponding to this penalty, which we denote $\mathcal{L}_{\text{LLP}}(T)$:
> $$
> \mathcal{L}_{\text{LLP}}(T) := \sum_j \sum_{i \in B_j} \sum_k T_{k,i} (1 - p(k \mid B_j)).
> $$
> Minimizing this term is equivalent to maximizing the alignment between the transport plan's implied label proportions and the true bag proportions. By defining the transport plan's label distribution in bag $B_j$ as $P_T(k \mid B_j)$, minimizing $\mathcal{L}_{\text{LLP}}(T)$ becomes equivalent to maximizing the sum of dot products between the predicted and true label distributions within each bag:
> $$
> \max_{T} \sum_j \langle P_T(\cdot \mid B_j), p(\cdot \mid B_j) \rangle.
> $$
> **Maximizing this dot product encourages the distribution $P_T$ to align with the true prior $p$**, which is analogous to minimizing statistical divergences like the KL divergence. Therefore, by introducing the LLP-Proportion Penalty into the cost matrix, we directly enforce that the optimal transport plan $T^*$—the unique solution to the optimization problem—favors assignments that are consistent with the known bag-level supervision.
>
>
>
> Reference:
>
> [1] Byrne, Charles L. Alternating minimization and alternating projection algorithms: A tutorial.
>
> # Weakness：more experiments are suggested.
>
>  We appreciate the feedback on the experimental scope. To address this, we have significantly expanded our evaluation:
> - **Beyond image datasets**: we have conducted additional experiments on tabular datasets to demonstrate the applicability of our method beyond image-based modalities. Specifically, we performed experiments on the UCI Adult dataset, a commonly used benchmark in tabular data. We also reproduced the results of two existing methods, LLP-AHIL and ROT, on this dataset for comparison. The experimental results are summarized in the table below:
> | Model | RLPL(ours) | LLP-AHIL | ROT |
> |:-----:|:----------:|:--------:|:---:|
> | UCI Adult | 77.57 | 75.99 | 72.82 |
>
> From the table, we observe that our method (RLPL) outperforms both LLP-AHIL and ROT on the UCI Adult dataset, achieving an accuracy of 77.57%. This indicates that our method is effective in the tabular data domain, achieving competitive performance compared to state-of-the-art methods. These results reinforce the robustness of our method across different non-vision modalities. We believe these experiments provide strong evidence that our approach generalizes well beyond image data and performs effectively in diverse domains such as tabular and structured data. In the final version of the paper, we will include these additional experiments and the corresponding analysis to further demonstrate the broad applicability of our method.
>
>
> # Question： Multi-class vs. Multi-label Setting
>
> Thank you for your comments. We give the following explanations.
>
> **1. Clarification of the Problem Setting:** Our work addresses the standard **multi-class** Learning from Label Proportions (LLP) problem, where each instance belongs to exactly one class from a set of $K$ mutually exclusive classes. Multiple samples form a bag, and we have the label distribution of the bag as supervision information. The problem is formally defined in Section 3, where the proportion for class $k$ in a bag $B_j$ is given by $p_{j,k} = \frac{1}{n_j} \sum_{i \in B_j} I(y_i = c_k)$, with $\sum_{k=1}^K p_{j,k} = 1$. This formulation inherently assumes a multi-class ground truth. Consequently, our entire framework, including the pseudo-labeling step, is designed for this multi-class setting. The use of `argmax` to generate a single pseudo-label per instance is the standard and appropriate procedure for deriving hard labels from a multi-class classifier's probability distribution.
>
> **2. Potential Source of Misunderstanding:** We believe the term "label proportions" itself, while standard in the LLP literature, might be interpreted in other machine learning contexts as relating to multi-label problems. We will ensure this distinction is made clearer in the final version of our paper to prevent future ambiguity.
>
> **3. On the Extension to a "Truly Multi-label" Setting:** We agree that extending this work to the **multi-label LLP** setting is a fascinating and important future research direction. Such an extension would require some significant modifications to our framework:
>
> - The classifier head would need to be changed from a `softmax` to multiple independent `sigmoid` activations.
> - The bag proportion loss would need to be redefined, as the sum of proportions would no longer be 1.
> - The OT formulation would be more complex, as the goal is no longer to assign an instance to a single class prototype. One might need to explore alternative matching frameworks.

---

> > ### Author Response · Authors · 2025-08-02
> > **Revision for format problem in "1. Existence and Uniqueness of the Optimal Solution”**
> >
> > Dear Reviewer oaZQ,
> >
> > we sincerely apologize for the formatting errors in our initial rebuttal, which made the equations and other parts difficult to read. We have corrected the formatting and are posting the full, properly-rendered response below for your convenience. Thank you for your patience and understanding.
> >
> > **1. Existence and Uniqueness of the Optimal Solution**
> > We first establish that the core optimization problem in our LLP-OTD mechanism has a unique global minimizer. The problem is defined as:
> > $$
> > \min_{T \in U(a, b)} \langle C, T \rangle - \gamma H(T),
> > $$
> > where $ \langle C, T \rangle = \sum_{k,i} C_{k,i} T_{k,i} $ is the transport cost, $H(T) = -\sum_{k,i} T_{k,i} (\log T_{k,i} - 1) $ is the entropy term, $\gamma > 0$ is the regularization strength, and $ U(a, b) $ is the transport polytope defined by marginal constraints.
> >
> > **Proof:**
> > Let $F(T) = \langle C, T \rangle - \gamma H(T)$. Then we will prove the strict convexity of the objective function. For the term $\langle C, T \rangle$, it is linear in $T$, hence convex. For the entropy term $-H(T) = \sum_{k,i} T_{k,i} (\log T_{k,i} - 1)$, we can compute its Hessian matrix:
> > $$
> > \frac{\partial^2 (-H)}{\partial T_{k,i}^2} = \frac{1}{T_{k,i}} > 0 \quad \text{since } T_{k,i} > 0,
> > $$
> > implying strict convexity of $-H(T)$. Therefore, $F(T) = \langle C, T \rangle - \gamma H(T)$ is the sum of a convex and strictly convex function, and thus strictly convex.
> >
> > Meanwhile, for
> > $ U(a,b) = \\left\\{ T \\in \\mathbb{R}^{K \\times N} \\, \\middle| \\, T_{k,i} \\ge 0, \\sum_i T_{k,i} = a_k, \\sum_k T_{k,i} = b_i \\right\\} $，, we will prove it to be convex, closed and bounded. As $U(a,b)$ is defined by linear equalities and nonnegativity constraints, it is a convex polytope. Because all constraints are equation or non-strict non-equation, $U(a,b)$ is closed. Since $\sum a_k = \sum b_i = 1$, we have$0 \le T_{k,i} \le 1$, and $U(a,b)$ is a compact convex set. Finally, by standard results in convex optimization, a strictly convex function over a non-empty compact convex set has a **unique global minimizer** $T^*$.

---

> > ### Author Response · Authors · 2025-08-02
> > **Revision for format problem in "2. Convergence of the Sinkhorn-Knopp Algorithm”**
> >
> > **2. Convergence of the Sinkhorn-Knopp Algorithm**
> > Next, we prove that the Sinkhorn-Knopp algorithm, used to solve the OT problem, converges to the unique optimal solution  $T^\*$ .
> > The optimal transport plan $T^\*$ is known to have a specific structure:
> > $$
> > T^*_{k,i} = u_k \cdot K_{k,i} \cdot v_i, \quad \text{where } K_{k,i} = \exp\left(-\frac{C_{k,i}}{\gamma}\right)
> > $$
> > The Sinkhorn-Knopp algorithm is an iterative procedure to find the scaling vectors $u \in \mathbb{R}^K$ and $v \in \mathbb{R}^N$ that satisfy the marginal constraints.
> >
> > **Proof：**
> > The convergence can be shown by interpreting the algorithm as an alternating projection procedure using the Kullback-Leibler (KL) divergence. Let the sets of matrices satisfying the row and column constraints be:
> > - $$ C_1 = \\left\\{ T \\in \\mathbb{R}^{K \\times N} \\middle| T_{k,i} \\ge 0, \\sum_i T_{k,i} = a_k \\right\\} $$ (Row constraints)
> > - $$ C_2 = \\left\\{ T \\in \\mathbb{R}^{K \\times N} \\middle| T_{k,i} \\ge 0, \\sum_k T_{k,i} = b_i \\right\\}$$ (Column constraints)
> >
> > Each iteration of the Sinkhorn-Knopp algorithm can be viewed as performing the following alternating KL projections:
> > $$
> > \begin{aligned}
> > T^{(l+\frac{1}{2})} &= \arg\min_{T \in C_1} \mathrm{KL}(T \parallel T^{(l)}), \\\\
> > T^{(l+1)} &= \arg\min_{T \in C_2} \mathrm{KL}(T \parallel T^{(l+\frac{1}{2})}),
> > \end{aligned}
> > $$
> > where the KL divergence is a specific type of Bregman divergence. Since $ C_1 $ and $ C_2$ are convex sets, the theory of alternating Bregman projections guarantees that this iterative process converges [1]. The intersection $ C_1 \cap C_2 =U(a,b)$ is non-empty and contains the unique optimal solution $T^\*$. Therefore, the sequence of iterates $T(l)$ produced by the Sinkhorn-Knopp algorithm is guaranteed to **converge to the unique optimal solution $T^*$**.

---

> > ### Author Response · Authors · 2025-08-02
> > **Revision for format problem in "3. Role of the LLP-Proportion Penalty”**
> >
> > **3. Role of the LLP-Proportion Penalty**
> > Finally, we provide the intuition for why our innovative LLP-Proportion Penalty helps align the pseudo-label distribution with the true LLP prior. Our effective cost function is:
> > $$
> > C_{\text{eff}}(c_k, z_i) = d(c_k, z_i) + \lambda_{\text{LLP}}(1 - p(k \mid B_j(i)))
> > $$
> > The key component is the penalty term. To understand its effect, we analyze the part of the objective function corresponding to this penalty, which we denote $L_{\text{LLP}}(T)$:
> > $$
> > L_{\text{LLP}}(T) := \sum_j \sum_{i \in B_j} \sum_k T_{k,i} (1 - p(k \mid B_j)).
> > $$
> > Minimizing this term is equivalent to maximizing the alignment between the transport plan's implied label proportions and the true bag proportions. By defining the transport plan's label distribution in bag $B_j$ as $P_T(k \mid B_j)$, minimizing $L_{\text{LLP}}(T)$ becomes equivalent to maximizing the sum of dot products between the predicted and true label distributions within each bag:
> > $$
> > \max_{T} \sum_j \langle P_T(\cdot \mid B_j),\ p(\cdot \mid B_j) \rangle.
> > $$
> > **Maximizing this dot product encourages the distribution $P_T$ to align with the true prior $p$**, which is analogous to minimizing statistical divergences like the KL divergence. Therefore, by introducing the LLP-Proportion Penalty into the cost matrix, we directly enforce that the optimal transport plan $T^*$—the unique solution to the optimization problem—favors assignments that are consistent with the known bag-level supervision.

---

> > > ### Author Response · Authors · 2025-08-04
> > >
> > > Dear Reviewer **oaZQ**
> > >
> > > We appreciate your valuable comments and positive recommendation. In the rebuttal, we have added **three theoretical guarantees**:
> > > 1. We theoretically explain why our proposed LLP-proportion penalty helps **align the pseudo-label distribution with the true LLP prior**.
> > > 2. We prove the **existence and uniqueness of the optimal solution** of our proposed LLP-OTD method.
> > > 3. We also prove the **convergence** of the adopted Sinkhorn-Knopp algorithm.
> > >
> > > Besides, we also **expand our experiments beyond the image datasets**.
> > > If there are any other questions or points you would like us to clarify further, we are more than willing to assist. Thank you again for your contribution to the review process.
> > >
> > > Regards.

---

> ### Author Response · Authors · 2025-08-06
>
> Dear  Reviewer **oaZQ**,
>
> We appreciate your valuable comments, which are important to improve the quality of this paper. We also appreciate your positive rating. **As the deadline for the author-reviewer discussion period is approaching**, please check whether your previous concerns have been fully addressed.
>
> Looking forward to your reply. Thanks.
>
> Regards.

---

> > ### Author Response · Authors · 2025-08-08
> >
> > Dear Reviewer **oaZQ**,
> >
> > Thanks again for your time and effort in reviewing this paper. We also appreciate your positive rating. As the author-reviewer discussion period will end in about one day, please check whether your previous concerns have been fully addressed. Thanks.
> >
> > Regards.

---

### Comment · Area_Chair_NDvR · 2025-08-05
**Author-Reviewer Discussion Phase**

Dear Authors and Reviewers,

Thank you for supporting NeurIPS 2025!

We are now in the final two days of the discussion phase. Please make the most of this time to exchange your views!

I encourage each Reviewer to read the Authors' responses and provide a reply, if you have not already done so.

Thank you,

NeurIPS 2025 Area Chair

---

### Note · Authors · 2025-08-13

Dear Senior Area Chair, Area Chair, and Reviewers,

We are sincerely grateful for the thorough review and constructive feedback our paper has received.

Our work introduces **RLPL**, a novel robust two-stage framework designed to tackle the critical challenge of pseudo-label noise in LLP. Its core innovations are **LLP-OTD**, which effectively refines pseudo-labels by embedding the LLP prior directly into the OT cost function, and **LLPMix**, which integrates a novel bag-level consistency loss into a semi-supervised pipeline, ensuring the final model adheres to LLP constraints, thereby achieving robust instance-level accuracy and proportion adherence.

We are delighted that through our detailed rebuttals and supplementary responses, we have successfully addressed all major concerns raised by the reviewers. This has led to a positive consensus, notably with Reviewers 3KP4 and 3xeg explicitly raising their scores after their concerns were successfully addressed. All reviewers have now provided supportive and positive ratings.

Following the reviewers' valuable suggestions, we have substantially strengthened our work by:
1. **Broadening experimental validation**: Added experiments on **tabular datasets** (UCI Adult, LLP-Bench) to demonstrate our method's cross-modality effectiveness.
2. **Enhancing robustness analysis**: Included new experiments under more challenging scenarios, such as **class imbalance and label proportion noise**, to validate our model's stability.
3. **Strengthening theoretical foundations**: Provided key **theoretical analysis** for our LLP-OTD method.
4. **Completing performance comparison**: Added a comprehensive comparison against **other strong and relevant methods (e.g., Count Loss, GLWS)** and analyzed computational efficiency.

We commit to integrating all these new experiments, analyses, and discussions into the final version of our paper.

Finally, we are sincerely grateful for the time and effort invested by the reviewers, the AC, and the SAC. Your dedication has been instrumental in improving our work.

Sincerely,

The Authors

---

### Decision · Program_Chairs · 2025-09-17

**Decision:**

Accept (poster)

**Comment:**

The paper concerns the problem of learning from label proportions (LLP). The Authors introduce a two-stage approach. In the first stage, they employ unsupervised contrastive learning and train an auxiliary classifier with bag-level supervision. In the second stage, they refine the obtained labels using an optimal-transport-based method. Finally, inspired by the MixMatch framework, they train the main classifier within a semi-supervised pipeline.

This is primarily an experimental paper. Nevertheless, the Reviewers appreciate the idea, which combines various concepts (contrastive learning, optimal transport, MixMatch) into a natural and "worth trying" approach for LLP problems. Most of the main concerns raised by the Reviewers were addressed in a satisfactory manner. While the paper lacks theoretical results, this weakness is outweighed by its other strengths.

Since NeurIPS is a very competitive conference, the final positive evaluation may not be sufficient for the paper to be accepted at this venue.